# Dissection of affinity captured LINE-1 macromolecular complexes

Martin S Taylor[1†], Ilya Altukhov[2†], Kelly R Molloy[3†], Paolo Mita[4], Hua Jiang[5], Emily M Adney[4,6], Aleksandra Wudzinska[6], Sana Badri[7], Dmitry Ischenko[2], George Eng[1], Kathleen H Burns[5,8], David Fenyö[4], Brian T Chait[3], Dmitry Alexeev[9], Michael P Rout[5], Jef D Boeke[4], John LaCava[4,5]*

[1]Department of Pathology, Massachusetts General Hospital, Boston, United States; [2]Moscow Institute of Physics and Technology, Dolgoprudny, Russia; [3]Laboratory of Mass Spectrometry and Gaseous Ion Chemistry, The Rockefeller University, New York, United States; [4]Department of Biochemistry and Molecular Pharmacology, Institute for Systems Genetics, NYU Langone Health, New York, United States; [5]Laboratory of Cellular and Structural Biology, The Rockefeller University, New York, United States; [6]McKusick-Nathans Institute of Genetic Medicine, Johns Hopkins University School of Medicine, Baltimore, United States; [7]Department of Pathology, NYU Langone Health, New York, United States; [8]Department of Pathology, Johns Hopkins University School of Medicine, Baltimore, United States; [9]Novosibirsk State University, Novosibirsk, Russia

*For correspondence:
jlacava@rockefeller.edu

[†]These authors contributed equally to this work

Competing interests: The authors declare that no competing interests exist.

**Abstract** Long Interspersed Nuclear Element-1 (LINE-1, L1) is a mobile genetic element active in human genomes. L1-encoded ORF1 and ORF2 proteins bind L1 RNAs, forming ribonucleoproteins (RNPs). These RNPs interact with diverse host proteins, some repressive and others required for the L1 lifecycle. Using differential affinity purifications, quantitative mass spectrometry, and next generation RNA sequencing, we have characterized the proteins and nucleic acids associated with distinctive, enzymatically active L1 macromolecular complexes. Among them, we describe a cytoplasmic intermediate that we hypothesize to be the canonical ORF1p/ORF2p/L1-RNA-containing RNP, and we describe a nuclear population containing ORF2p, but lacking ORF1p, which likely contains host factors participating in target-primed reverse transcription.
DOI: https://doi.org/10.7554/eLife.30094.001

## Introduction

Sequences resulting from retrotransposition constitute more than half of the human genome and are considered to be major change agents in eukaryotic genome evolution (*Kazazian, 2004*). L1 retrotransposons have been particularly active in mammals (*Furano et al., 2004*), comprising ~20% of the human genome (*Lander et al., 2001*); somatic retrotransposition has been widely implicated in cancer progression (*Lee et al., 2012*; *Tubio et al., 2014*) and may even play a role in neural development (*Muotri et al., 2005*). Despite the magnitude of their contributions to mammalian genomes, L1 genes are modest in size. A full-length L1 transcript is ~6 knt long and functions as a bicistronic mRNA that encodes two polypeptides, ORF1p and ORF2p (*Ostertag and Kazazian, 2001*), which respectively comprise a homotrimeric RNA binding protein with nucleic acid chaperone activity (*Martin and Bushman, 2001*) and a multifunctional protein with endonuclease and reverse transcriptase activities (*Mathias et al., 1991*; *Feng et al., 1996*). Recently, a putative primate-specific third ORF, named *ORF0*, has been identified on the Crick strand of the L1 gene; this ORF encodes a 71 amino acid peptide and may generate insertion-site-dependent ORFs via splicing (*Denli et al.,*

**eLife digest** Our genome consists of about two percent genes, while around 60 to 70 percent are made up of hundreds of thousands of copies of very similar DNA sequences. These repeats have accumulated over time due to specific genetic elements called transposons.

Transposons are often referred to as 'jumping genes', as they can move within the genome and thereby create mutations that may lead to cancer or other genetic diseases. LINE-1 is the only remaining active transposon in humans, and it expands by copying and pasting itself to new locations. To do so, it is first transcribed into RNA – the molecules that help to make proteins – and then converted back into identical DNA sequences.

In a never-ending battle, our cells have been fighting to keep LINE-1 and its ancestors from replicating, and so evolved various defense mechanisms. Yet, LINE-1 has learned to circumvent these barriers, and continues to replicate and cause disease. Our understanding of these defenses and of how LINE-1 evades them is limited.

Previous research has shown that the LINE-1 RNA and its two encoded proteins, called ORF1p and ORF2p, interact with a series of other proteins, with which they can form different types of complexes. Now, Taylor, Altukhov, Molloy et al. used human embryonic kidney cells grown in the laboratory with different LINE-1 mutations to identify how they affect the bound proteins and RNAs. The results showed that LINE-1 can form at least two different sets of complexes with other proteins.

The complex containing ORF1p and ORF2p and several other proteins was located in the cytoplasm, the fluid that fills the cells. However, the experiments also revealed a new complex in the cell nucleus, which contained ORF2p and proteins involved in DNA replication and repair, but not ORF1p. The results suggest ORF1p delivers RNPs to the nucleus around the time the cell divides. Another group of researchers has looked more closely at what happens during cell division.

A next step will be to study how exactly LINE-1 contributes to cancer. In the future, overactive LINE-1 proteins could be targeted to kill cancer cells, to identify cancer early, or to see if the cancer has come back. LINE-1 may also provide clues on how the genome has evolved.
DOI: https://doi.org/10.7554/eLife.30094.002

2015). ORF1p and ORF2p are thought to interact preferentially with the L1 RNA from which they were translated (in cis), forming a ribonucleoprotein (RNP) (*Kulpa and Moran, 2006*; *Taylor et al., 2013*) considered to be the canonical direct intermediate of retrotransposition (*Hohjoh and Singer, 1996*; *Kulpa and Moran, 2005*; *Martin, 1991*; *Kulpa and Moran, 2006*; *Doucet et al., 2010*). L1 RNPs also require host factors to complete their lifecycle (*Suzuki et al., 2009*; *Peddigari et al., 2013*; *Dai et al., 2012*; *Taylor et al., 2013*) and, consistent with a fundamentally parasitic relationship (*Beauregard et al., 2008*), the host has responded by evolving mechanisms that suppress retrotransposition (*Goodier et al., 2013*; *Arjan-Odedra et al., 2012*; *Goodier et al., 2012*; *Niewiadomska et al., 2007*). It follows that as the host and the parasite compete, L1 expression is likely to produce a multiplicity of RNP forms engaged in discrete stages of retrotransposition, suppression, or degradation.

Although L1 DNA sequences are modestly sized compared to typical human genes, L1 intermediates are nevertheless RNPs with a substantially sized RNA component; e.g. larger than the ~5 knt 28S rRNA (*Gonzalez et al., 1985*) and approximately three to four times the size of a 'typical' mRNA transcript (*Lander et al., 2001*; *Sommer and Cohen, 1980*). Therefore, it is likely that many proteins within L1 RNPs form interactions influenced directly and indirectly by physical contacts with the L1 RNA. We previously reported that L1 RNA comprised an estimated ~25% of mapped RNA sequencing reads in ORF2p-3xFLAG affinity captured fractions (*Taylor et al., 2013*). We also observed that the retention of ORF1p and UPF1 within affinity captured L1 RNPs was reduced by treatment with RNases (*Taylor et al., 2013*). In the same study we observed that two populations of ORF2p-associated proteins could be separated by split-tandem affinity capture (ORF2p followed by ORF1p), a two-dimensional affinity enrichment procedure (*Caspary et al., 1999*; *Taylor et al., 2013*). Initial characterization of these two L1 populations by western blotting suggested that discrete L1 populations were likely primed for function in different stages of the lifecycle. We therefore

expected additional uncharacterized complexity in the spectrum of L1-associated complexes present in our affinity enriched fractions.

In this study, we have used quantitative mass spectrometry (MS) to investigate the proteomic characteristics of endogenously assembled ectopic L1-derived macromolecules present in an assortment of affinity-enriched fractions. We revisited RNase treatment and split-tandem affinity capture approaches and complemented them with RNA sequencing, enzymatic analysis, and in-cell localization of ORF proteins by immunofluorescence microscopy (see also the companion manuscript by *Mita et al., 2018*). We additionally explored proteomes associated with catalytically-inactivated ORF2p point mutants and monitored the rates of protein exchange from L1 macromolecules in vitro. Taken together, our data support the existence of a variety of putative L1-related protein complexes.

## Results

Affinity proteomic experiments conducted in this study use quantitative MS based upon metabolic labeling (*Oda et al., 1999*). Two main experimental designs (and modifications thereof) facilitating quantitative cross-sample comparisons have been used: SILAC (*Ong et al., 2002*; *Wang and Huang, 2008*) and I-DIRT (*Tackett et al., 2005*; *Taylor et al., 2013*). In these approaches, cells are grown for several doublings in media containing amino acids composed either of naturally-occurring 'light' isotopes or biologically identical 'heavy' isotopes (e.g. $^{13}$C, $^{15}$N lysine and arginine), such that the proteomes are thoroughly labeled. Protein fractions derived from the differently labeled cell populations, obtained e.g. before and after experimental manipulations are applied, are mixed and the relative differences in proteins contributed by each fraction are precisely measured by mass spectrometry. In addition to the above cited studies, these approaches have been adapted to numerous biological questions using a variety of analytical frameworks e.g. (*Byrum et al., 2011*; *Luo et al., 2016*; *Trinkle-Mulcahy et al., 2008*; *Ohta et al., 2010*; *Kaake et al., 2010*; *Geiger et al., 2011*). Because it is challenging to speculate on the potential physiological roles of protein interactions that form after extraction from the cell, we often use I-DIRT, which allows the discrimination of protein-protein interactions formed in-cell from those occurring post-extraction. Our prior affinity proteomic study, based on I-DIRT, identified 37 putative *in vivo* interactors (*Taylor et al., 2013*), described in *Table 1*. In this study we primarily analyze the behaviors of these 'I-DIRT significant' L1 interactors, in order to determine their molecular associations and ascertain the variety of distinctive macromolecular complexes formed in-cell that copurify with affinity-tagged ORF2p. The complete lists of proteins detected in each experiment are presented in the supplementary information (see *Supplementary file 1*). We have represented any ambiguous protein group, which occurs when the same peptides identify a group of homologous protein sequences, with a single, consistently applied gene symbol and a superscript 'a' in all figures. *Supplementary file 1* contains the references to other proteins explaining the presence of the same peptides. For example, RPS27A, (ubiquitin) UBB, UBC, and (ribosomal Protein L40) UBA52 can be explained by common ubiquitin peptides shared by these genes. RPS27A-specific peptides were not identified in this study, but we retained the nomenclature for consistency with our previous work; HSPA1A is reported in this study, but cannot be distinguished from the essentially identical protein product of HSPA1B.

Except where noted otherwise, the presented experiments were conducted in suspension-cultured HEK-293T$_{LD}$ cells, using a synthetic L1 construct - *ORFeus*-HS - driving the expression 3xFLAG-tagged L1 (*ORF1; ORF2::3xFLAG*; 3'-UTR) from a tetracycline inducible minimal-CMV promoter, harbored on a mammalian episome (pLD401 (*Taylor et al., 2013*; *An et al., 2011*; *Dai et al., 2012*)). All L1-related macromolecules described in this study were obtained by affinity capture of ORF2p-3xFLAG before further experimental manipulations were applied. We consider macromolecules containing L1 RNA (L1 RNPs, discussed throughout) and/or an L1 cDNA (i.e. L1 coding potential) to be L1s, as are their ectopic plasmid-borne and endogenous gDNA counterparts, reflecting the complexity and diversity of L1 forms arising from its lifecycle. In an effort to characterize this complexity, we have carried out RNA sequencing and enzymatic activity analyses on several affinity captured fractions, complementing the proteomic analyses.

**Table 1.** Putative L1 interactors: Through a series of affinity capture experiments (co-IP) using I-DIRT, we characterized a set of putative host-encoded L1 interactors (**Taylor et al., 2013**).
The proteins observed were associated with both ORF1p and ORF2p (highlighted in blue), only in association with ORF2p (highlighted in magenta), or only in association with ORF1p (no highlight). The two highlighted populations are the central focus of this study.

| Gene symbol | Uniprot symbol | Protein | co-IP with |
| --- | --- | --- | --- |
| L1RE1 | Q9UN81 | ORF1p | ORF1/2 |
| N/A | O00370 | ORF2p | ORF1/2 |
| MOV10 | Q9HCE1 | Putative helicase MOV-10 | ORF1/2 |
| PABPC1 | P11940 | Polyadenylate-binding protein 1 | ORF1/2 |
| PABPC4 | Q13310 | Polyadenylate-binding protein 4 | ORF1/2 |
| UPF1 | Q92900 | Regulator of nonsense transcripts 1 | ORF1/2 |
| ZCCHC3 | Q9NUD5 | Zinc finger CCHC domain-containing protein 3 | ORF1/2 |
| FKBP4 | Q02790 | Peptidyl-prolyl cis-trans isomerase FKBP4 | ORF2 |
| HAX1 | O00165 | HCLS1-associated protein X-1 | ORF2 |
| HMCES | Q96FZ2 | Embryonic stem cell-specific 5-hydroxymethylcytosine-binding protein | ORF2 |
| HSP90AA1 | P07900 | Heat shock protein HSP 90-alpha | ORF2 |
| HSP90AB1 | P08238 | Heat shock protein HSP 90-beta | ORF2 |
| HSPA1A | P0DMV8 | Heat shock 70 kDa protein 1A | ORF2 |
| HSPA8 | P11142 | Heat shock cognate 71 kDa protein | ORF2 |
| IPO7 | O95373 | Importin-7 | ORF2 |
| NAP1L1 | P55209 | Nucleosome assembly protein 1-like 1 | ORF2 |
| NAP1L4 | Q99733 | Nucleosome assembly protein 1-like 4 | ORF2 |
| PARP1 | P09874 | Poly [ADP-ribose] polymerase 1 | ORF2 |
| PCNA | P12004 | Proliferating cell nuclear antigen | ORF2 |
| PURA | Q00577 | Transcriptional activator protein Pur-alpha | ORF2 |
| PURB | Q96QR8 | Transcriptional activator protein Pur-beta | ORF2 |
| RPS27A | P62979 | Ubiquitin-40S ribosomal protein S27a | ORF2 |
| TIMM13 | Q9Y5L4 | Mitochondrial import inner membrane translocase subunit Tim13 | ORF2 |
| TOP1 | P11387 | DNA topoisomerase 1 | ORF2 |
| TOMM40 | O96008 | Mitochondrial import receptor subunit TOM40 homolog | ORF2 |
| TUBB | P07437 | Tubulin beta chain | ORF2 |
| TUBB4B | P68371 | Tubulin beta-4B chain | ORF2 |
| YME1L | Q96TA2 | ATP-dependent zinc metalloprotease YME1L1 | ORF2 |
| CORO1B | Q9BR76 | Coronin-1B | ORF1 |
| DDX6 | P26196 | Probable ATP-dependent RNA helicase DDX6 | ORF1 |
| ERAL1 | O75616 | GTPase Era, mitochondrial | ORF1 |
| HIST1H2BO | P23527 | Histone H2B type 1-O | ORF1 |
| LARP7 | Q4G0J3 | La-related protein 7 | ORF1 |
| MEPCE | Q7L2J0 | 7SK snRNA methylphosphate capping enzyme | ORF1 |
| PABPC4L | P0CB38 | Polyadenylate-binding protein 4-like | ORF1 |
| TROVE2 | P10155 | 60 kDa SS-A/Ro ribonucleoprotein | ORF1 |
| YARS2 | Q9Y2Z4 | Tyrosine–tRNA ligase, mitochondrial | ORF1 |

DOI: https://doi.org/10.7554/eLife.30094.003

## RNase-sensitivity exhibited by components of affinity captured L1 RNPs

*Figure 1* (panels A-C) illustrates the approach and displays the findings of our assay designed to reveal which proteins depend upon the presence of intact L1 RNA for retention within the obtained L1 RNPs. Briefly, metabolically-labeled affinity captured L1s were treated either with a mixture of RNases A and T1 — thus releasing proteins that require intact RNA to remain linked to ORF2p and the affinity medium — or BSA, as an inert control. After removing the fractions released by the RNase or BSA treatments, the proteins remaining on the affinity media were eluted with lithium dodecyl sulfate (LDS), mixed together, and then analyzed by MS. Proteins released, and so depleted, by RNase treatment were thus found to be more abundant in the BSA-treated control. The results obtained corroborate and extend our previous findings: ORF1p and UPF1 exhibited RNase-sensitivity (*Taylor et al., 2013*). We also observed that ZCCHC3 and MOV10 exhibited RNase-sensitivity to a level similar to ORF1p. The remaining I-DIRT significant proteins were RNase-resistant in this assay. With the exception of the PABPC1/4 proteins (and ORF2p itself, see Discussion), the I-DIRT significant proteins (colored nodes, *Figure 1C*) that were resistant to RNase treatment (nearest the origin of the graph) classify ontologically as nuclear proteins (GO:0005634, $p \approx 3 \times 10^{-4}$, see Materials and methods). These same proteins were previously observed as specific L1 interactors in I-DIRT experiments targeting ORF2p but not in those targeting ORF1p; in contrast, the proteins that demonstrated RNase-sensitivity: ORF1, MOV10, ZCCHC3, and UPF1 were observed in both ORF1p and ORF2p I-DIRT experiments (*Table 1*). Stated another way, the proteins released upon treating an affinity captured ORF2p fraction with RNases are among those that can also be obtained when affinity capturing ORF1p directly, while those that are RNase-resistant are not ORF1p interactors (*Taylor et al., 2013*). The ORF1p-linked, I-DIRT significant, RNase-sensitive proteins were too few to obtain a high confidence assessment of ontological enrichment; but, when combined with the remaining proteins exhibiting sensitivity to RNase treatment (black nodes, *Figure 1C*), they together classified as 'RNA binding' (GO:0003723, $p \approx 1 \times 10^{-11}$). This analysis also revealed a statistically significant overrepresentation of genes associated with the exon junction complex (EJC, GO: 0035145, $p \approx 1 \times 10^{-6}$, discussed below). Hence, the overlapping portion of the ORF1p- and ORF2p-associated interactomes appeared to depend upon intact L1 RNA. Host-encoded proteins segregated into groups that responded differentially to RNase treatment, with a substantial population of RNase-resistant interactors linked to both ORF2p and the nucleus. This observation led to the hypothesis that our ORF2p-3xFLAG affinity captured L1s constitute a composite purification of at least, but not limited to, (1) a population of L1-RNA-dependent, ORF1p/ORF2p-containing L1 RNPs, and (2) an ORF1p-independent nuclear population associated with ORF2p.

While effects of PABPC1, MOV10, and UPF1 on L1 activity have been described (*Arjan-Odedra et al., 2012*; *Taylor et al., 2013*; *Dai et al., 2012*), effects of ZCCHC3 on L1 remained uncharacterized. ZCCHC3 is an RNA-binding protein associated with poly(A)+ RNAs (*Castello et al., 2012*) but otherwise little is known concerning its functions. Notably, in a genome-wide screen, small interfering (si)RNA knockdown of ZCCHC3 was observed to increase the infectivity of the Hepatitis C, a positive sense RNA virus (*Li et al., 2009*); and ZCCHC3 was observed to copurify with affinity captured HIV, a retrovirus, at a very high SILAC ratio (>10), supporting the specificity of this interaction (*Engeland et al., 2014*). We therefore explored the effects on L1 mobility both of over-expression and siRNA knockdown of ZCCHC3. Over-expression of ZCCHC3 reduced L1 retrotransposition to ~10% that observed in the control, consistent with a negative regulatory role for ZCCHC3 in the L1 lifecycle; small interfering RNA (siRNA) knockdown of ZCCHC3 induced a modest increase in retrotransposition compared to a scrambled control siRNA (~1.9x ± 0.1; *Supplementary file 2*). Moreover, although not among our I-DIRT hits (see Discussion), the presence of EJC components (MAGOH, RBM8A, EIF4A3, UPF1) among the RNase-sensitive fraction of proteins intrigued us, given that L1 genes are intronless. We speculated that L1s may use EJCs to enhance nuclear export, evade degradation by host defenses, and/or aggregate with mRNPs within cytoplasmic granules. For this reason we carried out a series of siRNA knockdowns of these EJC components and other physically or functionally related proteins found in the affinity captured fraction (listed in *Supplementary file 2*). siRNA knockdowns of RBM8A and EIF4A3 caused inviability of the cell line. We found that knocking-down MAGOH or the EJC-linked protein IGF2BP1 (*Jønson et al., 2007*) reduced

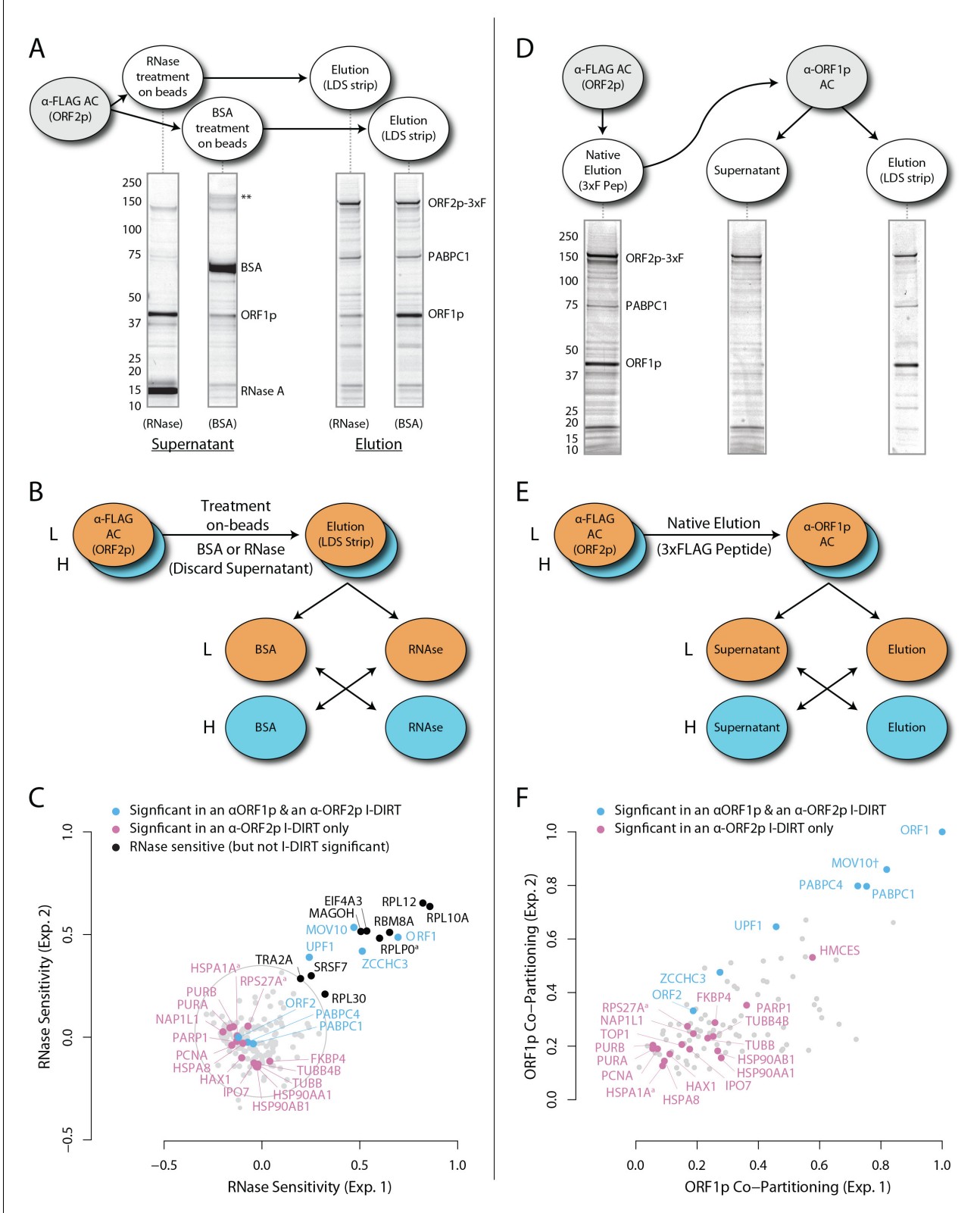

**Figure 1.** RNase sensitivity and split-tandem affinity capture of L1 ORF2p RNPs. (**A**) On-bead RNase-sensitivity assay: L1 complexes were affinity captured by ORF2p-3xFLAG. The magnetic media were then treated with a solution containing either a mixture of RNases A and T1 or BSA. After treatment, the supernatants were removed and the remaining bound material was released with LDS. Proteins requiring intact RNA to maintain stable interactions with immobilized ORF2p were released from the RNase-treated medium, while the BSA-treated sample controlled for the spontaneous

*Figure 1 continued on next page*

Figure 1 continued

release of proteins from the medium. Representative SDS-PAGE/Coomassie blue stained gel lanes are shown for each fraction. (B) The experiments described above were carried out in duplicate, once with light isotopically labeled cells (L) and once with heavy isotopically labeled cells (H), resulting in four label-swapped, SILAC duplicates (one light set and one heavy set). The four fractions were cross-mixed and the differential protein retention upon the affinity medium during the treatments (BSA vs. RNase) was assessed by quantitative MS. (C) Results from the RNase-sensitivity assay graphed as the fraction of each detected protein present in the BSA-treated sample (RNase-sensitive proteins are *more* present in the BSA treated sample), normalized such that proteins that did not change upon treatment with RNases are centered at the origin. A cut-off of $p=10^{-3}$ for RNase-sensitivity is indicated by a light gray circle; proteins that are RNase-sensitive with a statistical significance of $p<10^{-3}$ are outside the circle. Proteins previously ranked significant by I-DIRT analysis (*Table 1*) are labeled and displayed in blue or magenta (as indicated); black nodes were RNase-sensitive but not significant by I-DIRT; gray, unlabeled nodes were neither RNase-sensitive nor significant by I-DIRT. (D) *Split-tandem affinity capture*: L1 complexes were affinity captured by ORF2p-3xFLAG. After native elution with 3xFLAG peptide, this fraction was depleted of ORF1p-containing complexes using an α-ORF1 conjugated magnetic medium, resulting in a supernatant fraction depleted of ORF1p-containing complexes. The α-ORF1 bound material was then released with LDS, yielding an elution fraction enriched for ORF1p-containing complexes. Representative SDS-PAGE/Coomassie blue stained results for each fraction are shown. (E) SILAC duplicates, two supernatants and two elutions, were cross-mixed to enable an assessment of the relative protein content of each fraction by quantitative MS. (F) The results from split-tandem affinity capture graphed as the fraction of each protein observed in the elution sample. In order to easily visualize the relative degree of co-partitioning of constituent proteins with ORF1p, these data were normalized, setting the fraction of ORF1p in the elution to 1. Proteins which were previously ranked significant by I-DIRT analysis are labeled and displayed in blue or magenta (as indicated); gray, unlabeled nodes were not found to be significant by I-DIRT. MOV10 is marked with a dagger because in one replicate of this experiment it was detected by a single unique peptide, whereas we have enforced a minimum of two peptides (see Materials and methods) for all other proteins, throughout all other proteomic analyses presented here.

DOI: https://doi.org/10.7554/eLife.30094.004

retrotransposition by ~50%, consistent with a role in L1 proliferation; although these knockdowns also caused a reduction in viability of the cell line (see Discussion).

## Split-tandem separation of compartment-specific L1 ORF-associated complexes

To further test our hypothesis and better characterize the components of our L1 fraction, we conducted split-tandem affinity capture. *Figure 1* (panels D-F) illustrates the approach and displays the findings of the assay, which physically separated ORF1p/ORF2p-containing L1 RNPs from a presumptive 'only-ORF2p-associated' population. Briefly, metabolically-labeled L1s were affinity captured by ORF2p-3xFLAG (first dimension) and the obtained composite was subsequently further fractionated by α-ORF1p affinity capture (second dimension, or split-tandem capture), resulting in α-ORF1p-bound and unbound (supernatant) fractions. The bound fraction was eluted from the affinity medium with LDS (elution). The supernatant and elution fractions were then mixed and analyzed by MS to ascertain proteomic differences between them. The α-ORF1p elution contained the population of proteins physically linked to *both* ORF2p and ORF1p, whereas the supernatant contained the proteins associated *only* with ORF2p (and, formally, those which have dissociated from the ORF1p/ORF2p RNP). The results corroborated our previous observations that: (i) almost all of the ORF1p partitioned into the elution fractions, (ii) a quarter of the ORF2p (~26%) followed ORF1p during the α-ORF1p affinity capture, (iii) roughly half of the UPF1 (~55%) followed ORF1p, and (iv) most of the PCNA (~87%) remained in the ORF1p-depleted supernatant fraction (*Figure 1F*, and consistent with prior estimates based on protein staining and western blotting [*Taylor et al., 2013*]); thus (v) supporting the existence of at least two distinct populations of L1-ORF-protein-containing complexes in our affinity purifications.

The population eluted from the α-ORF1p affinity medium (*Figure 1D*, far right gel lane, and nodes located in the upper right of the graph, panel F) is consistent with the composition of the ORF1p/ORF2p-containing L1 RNP suggested above. Our split-tandem separation segregated the constituents of the L1 fraction comparably to the RNase-sensitivity assay, both in terms of which proteins co-segregated with ORF1p/ORF2p (compare *Figure 1C and F*, blue nodes, upper right of graphs) as well as those which appear to be linked only to ORF2p (compare *Figure 1C and F*, magenta nodes, lower left of the graphs). The ORF1p/ORF2p RNPs obtained by split-tandem capture included putative in vivo interactions associated with both α-ORF1p and α-ORF2p I-DIRT affinity capture experiments; whereas the unbound, ORF1p-independent fraction includes proteins previously observed as significant only in α-ORF2p I-DIRT experiments (*Table 1*). Analysis of the nodes whose degree of ORF1p association was similar to that of UPF1 (blue nodes exhibiting ≥55%

ORF1p co-partitioning, *Figure 1F*) revealed that they map ontologically to a 'cytoplasmic ribonucleoprotein granule' classification (GO:0036464, p $\approx$ 6 $\times$ 10$^{-8}$; see Discussion). In contrast, all sixteen proteins exhibiting ORF1p co-partitioning approximately equal to or less than that of ORF2p were predominantly found in the supernatant fraction and were enriched for cell-compartment-specific association with the nucleus (GO:0005634, p $\approx$ 4 $\times$ 10$^{-5}$; *Figure 1F*: all magenta nodes $\leq$36%). These two fractions therefore appear to be associated with different cell compartments, reaffirming our postulate: the ORF1p/ORF2p-containing population is a cytoplasmic intermediate related to the canonical L1 RNP typically ascribed to L1 assembly in the literature, and the predominantly ORF2p-associated population comprises a putative nuclear interactome.

From the same analysis, we noted that PURA, PURB, PCNA, and TOP1 which all partition predominantly with nuclear L1, exhibited an ontological co-enrichment (termed 'nuclear replication fork,' GO:0043596, p $\approx$ 3 $\times$ 10$^{-4}$). The nodes representative of PURA, PURB, and PCNA appeared to exhibit a striking proximity to one another, suggesting highly similar co-fractionation behavior potentially indicative of direct physical interactions. In an effort to examine this possibility, we graphed the frequency distribution of the proximities of all three-node-clusters observed within *Figure 1F*, revealing the likelihood of the PURA/PURB/PCNA cluster to be p=3.2$\times$10$^{-7}$ (see Appendix 1). We therefore concluded that PURA, PURB, PCNA, and (perhaps at a lower affinity) TOP1, likely constitute a physically associated functional module interacting with L1. In further support of this assertion, we noted that known functionally linked protein pairs PABPC1/PABPC4 (cytoplasmic) (*Jønson et al., 2007*; *Katzenellenbogen et al., 2007*) and HSPA8/HSPA1A (nuclear) (*Jønson et al., 2007*; *Nellist et al., 2005*) also exhibited comparable co-partitioning by visual inspection, and statistical testing of these clusters revealed the similarity of their co-partitioning to be significant at p $\approx$ 0.001 for the former, and p $\approx$ 0.0002 for the latter. The observed variation in co-partitioning behavior between the different proteins comprising the nuclear L1 fraction might reflect the presence of multiple distinctive (sub)complexes present within this population.

To validate our hypothesis that these proteins are associated with ORF2p in the nucleus, possibly engaged with host genomic DNA, we carried out ORF2p-3xFLAG affinity capture from chromatin-enriched sub-cellular fractions and found that the co-captured proteins we identified (*Supplementary file 3*) overlapped with those described above as nuclear interactors, including: PARP1, PCNA, UPF1, PURA, and TOP1. We previously demonstrated that silencing PCNA expression adversely affects L1 retrotransposition (*Taylor et al., 2013*), in this study we found that knocking down TOP1 approximately doubled retrotransposition frequency, while a more modest 1.4x increase effect was observed for PURA, and no substantial effect was observed for PURB, compared to a scrambled siRNA control. In contrast, over-expression of PURA reduced retrotransposition to ~20% of the expected level (*Supplementary file 2*). IPO7 was also observed among the putative ORF2p co-factors within the chromatin enriched fraction, congruent with its matching behavior in *Figure 1C and F*. Notably, IPO7 functions as a nuclear import adapter for HIV reverse transcription complexes (*Fassati et al., 2003*). Several other proteins that were observed did not previously exhibit I-DIRT specificity (*Supplementary file 3*).

## L1 RNA and LEAP activity in affinity captured fractions

Because the L1 RNA is an integral component of proliferating L1s, and because we observed that interactions between ORF2p, ORF1p, and some host proteins were sensitive to treatment with RNases, we sought to characterize the RNAs present in our samples. We extracted RNAs from each of the three fractions produced by split-tandem affinity capture (*Figure 1D*) and carried out RNA sequencing; *Figure 2A* displays the sequence coverage observed across the entirety of our synthetic L1 construct in each fraction, revealing a normalized ~2 fold difference in abundance between the elution and supernatant fractions. Synthetic L1s constituted ~60% of the mapped, annotated sequence reads in the fractions eluted from the $\alpha$-FLAG and $\alpha$-ORF1p affinity media, and ~30% of the reads in the ORF1p-depleted supernatant fraction; sequencing reads mapping to protein coding genes made up the majority of the remaining annotated population in all fractions. We observed that a substantial number of reads mapped to unannotated regions of the human genome, in particular in the supernatant fraction, enriched for putative nuclear L1 complexes; the breakdown of mapped and annotated sequencing reads is summarized in *Figure 2B* and expanded in *Supplementary file 4*.

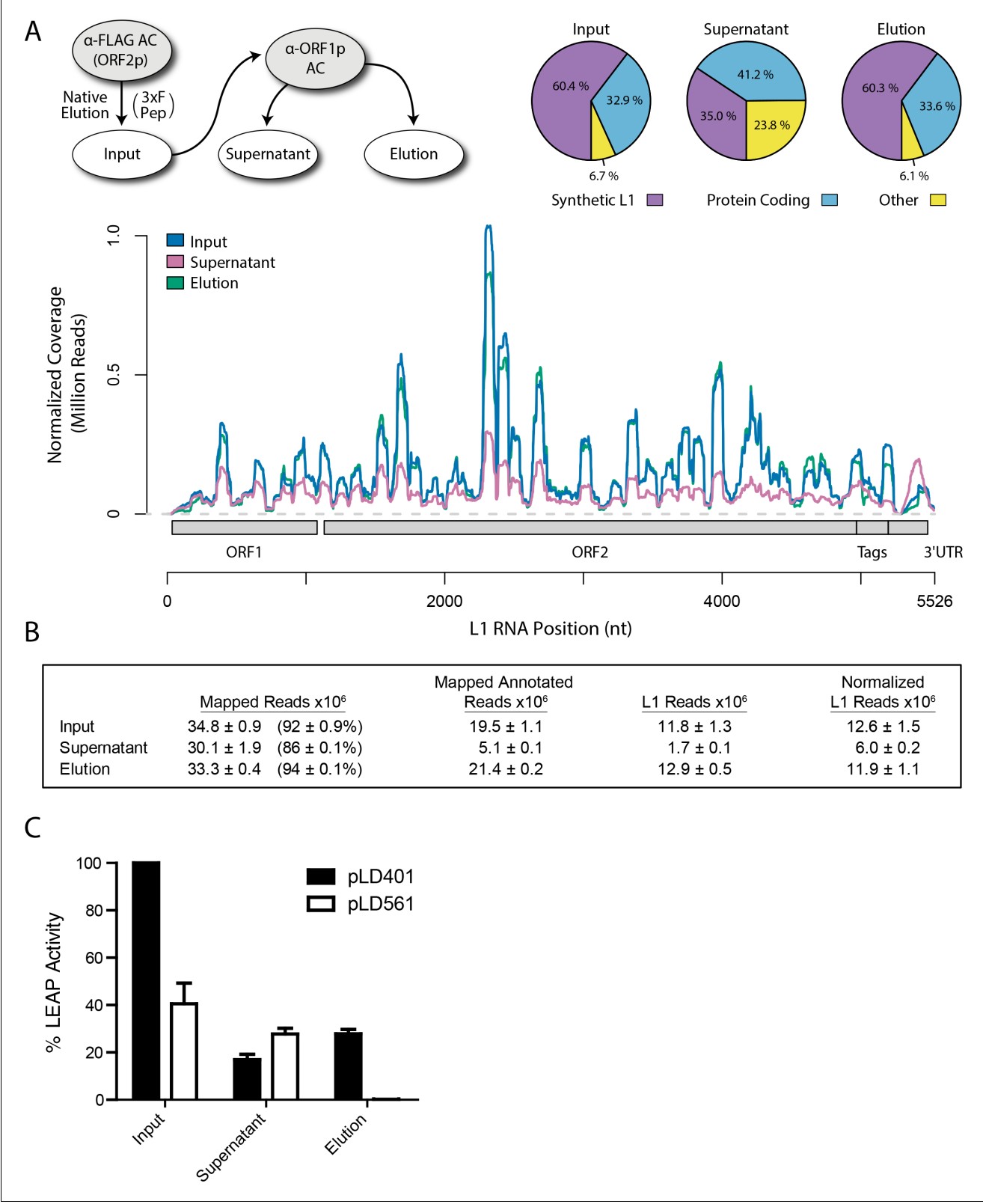

**Figure 2.** Transcriptomic and enzymatic analysis of split-tandem RNP fractions. (**A**) *RNA sequencing affinity captured L1s:* L1 complexes were obtained by split-tandem affinity capture, as in *Figure 1D* (simplified schematic shown); RNA extracted from these three fractions was subjected to next-generation sequencing. The results are summarized with respect to coverage of the synthetic L1 sequence (see schematic with nucleotide coordinates) as well as the relative quantities of mapped, annotated reads (pie charts; the mean of duplicate experiments is displayed). (**B**) *Summary of sequencing*

*Figure 2 continued on next page*

*Figure 2 continued*

reads: displays the total number of sequencing reads that mapped to our reference library, the subset of mapped reads carrying a genome annotation, and the number of reads that corresponding to L1, both raw and normalized (see Materials and methods and *Supplementary file 4*). The mean of duplicate experiments is displayed; ±indicates the data range. (C) *LINE-1 element amplification protocol (LEAP) of affinity captured L1s*: L1 complexes were obtained from full length synthetic L1 (pLD401) and an otherwise identical ΔORF1 construct (pLD561) following the same experimental design as in (A), except that elution from α-ORF1p affinity medium was done natively, by competitive elution. In this assay, L1 cDNAs are produced, in cis, by ORF2p catalyzed reverse transcription of L1 RNAs; the resulting cDNAs by were measured by quantitative PCR and presented as relative quantities normalized to pLD401 input (*Supplementary file 4*). The mean of duplicate experiments is displayed; error bars indicate the data range.

DOI: https://doi.org/10.7554/eLife.30094.005

Retrotransposition-competent L1 RNPs form in cis, with ORF proteins binding to the L1 RNA that encoded them ('*cis* preference'), presumably at the site of translation in the cytoplasm (*Kulpa and Moran, 2006*; *Wei et al., 2001*). Given that ORF1/2p partitioned to the split-tandem elution fraction (cytoplasmic) along with the greater fraction of L1 RNA, yet only ORF2p and a lesser portion of the L1 RNA were observed in the supernatant (nuclear), an important consideration regarding these fractions is: to what extent they contain L1 macromolecules capable of proliferation. To address this question, we performed the LINE-1 element amplification protocol (LEAP) on split-tandem affinity captured fractions (*Figure 2C*; *Supplementary file 4*), including a ΔORF1 construct (pLD561) as a control (*Taylor et al., 2013*). LEAP is currently the best biochemical assay for functional co-assembly of L1 RNA and proteins (*Kulpa and Moran, 2006*); it measures the ability of ORF2p to amplify its associated L1 RNA by reverse transcription. To execute LEAP on the α-ORF1p affinity captured fraction, we developed a competitive di-peptide elution reagent based on the linear peptide sequence used to generate the α-ORF1p 4H1 monoclonal antibody: residues 35–44 in ORF1p ([*Khazina et al., 2011*; *Taylor et al., 2013*]; see Materials and methods). We were thus able to assay the partitioning of enzymatic activity within the different populations of copurifying proteins in a split-tandem affinity capture experiment. Our data showed robust LEAP activity in both nuclear and cytoplasmic split-tandem supernatant and elution fractions. We note that our 3xFLAG eluted fractions have been shown to possess ~70 fold higher specific activity than L1 RNPs obtained by sucrose cushion velocity sedimentation (*Taylor et al., 2013*), hence the activity levels detected far exceed those obtained by sedimentation.

## ORF1p/ORF2p immunofluorescence protein localization

Although our proteomic and biochemical analyses supported the existence of distinctive nuclear and cytoplasmic L1 populations, our prior immunofluorescence (IF) analyses did not reveal an apparent nuclear population, leading us to revisit IF studies. Previously, IF of ORF1p and ORF2p in HeLa and HEK-293T cells yielded two striking observations: (i) ORF2 expression was seemingly stochastic, with ORF2p observed in ~30% of cells; and (ii) while ORF1p and ORF2p co-localized in cells that exhibited both, we did not observe an apparent nuclear population of either protein (*Taylor et al., 2013*). Subsequently, we noted an absence of mitotic cells from these preparations. Reasoning that these cells were lost due to selective adherence on glass slides, and noting that cell division has been reported to promote L1 transposition (*Xie et al., 2013*; *Shi et al., 2007*), we repeated the assays using puromycin-selected Tet-on HeLa cells grown on fibronectin coated coverslips. The results are shown in *Figure 3*.

The modified IF assay corroborated our prior results in that nearly all the cells exhibited cytoplasmic ORF1p and a minority subset of ~1/3rd also exhibited co-localized cytoplasmic ORF2p (*Figure 3A*, top row). We also observed an uncommon and previously unrecognized subpopulation of cells, consisting of pairs exhibiting nuclear localized ORF2p (*Figure 3A*, middle row); because these cells occurred in proximal pairs, we presumed them to have recently gone through mitosis. Statistical analysis of microscopy images displaying cells with nuclear localized ORF2p confirmed their proximities to be significantly closer than those of randomly selected cells (*Figure 3B*; *Supplementary file 5*). Expression of *ORF2* in the absence of *ORF1* (ΔORF1; pLD561) resulted in the majority of cells exhibiting cytoplasmic ORF2p, consistent with our previous work (*Taylor et al., 2013*). We did not observe instances of nuclear ORF2p using the ΔORF1 construct (*Figure 3A*, bottom row), suggesting that ORF1p is required for ORF2p nuclear localization (see Discussion). In a separate study, including more detailed analyses of ORF protein localization, *Mita et al., 2018*

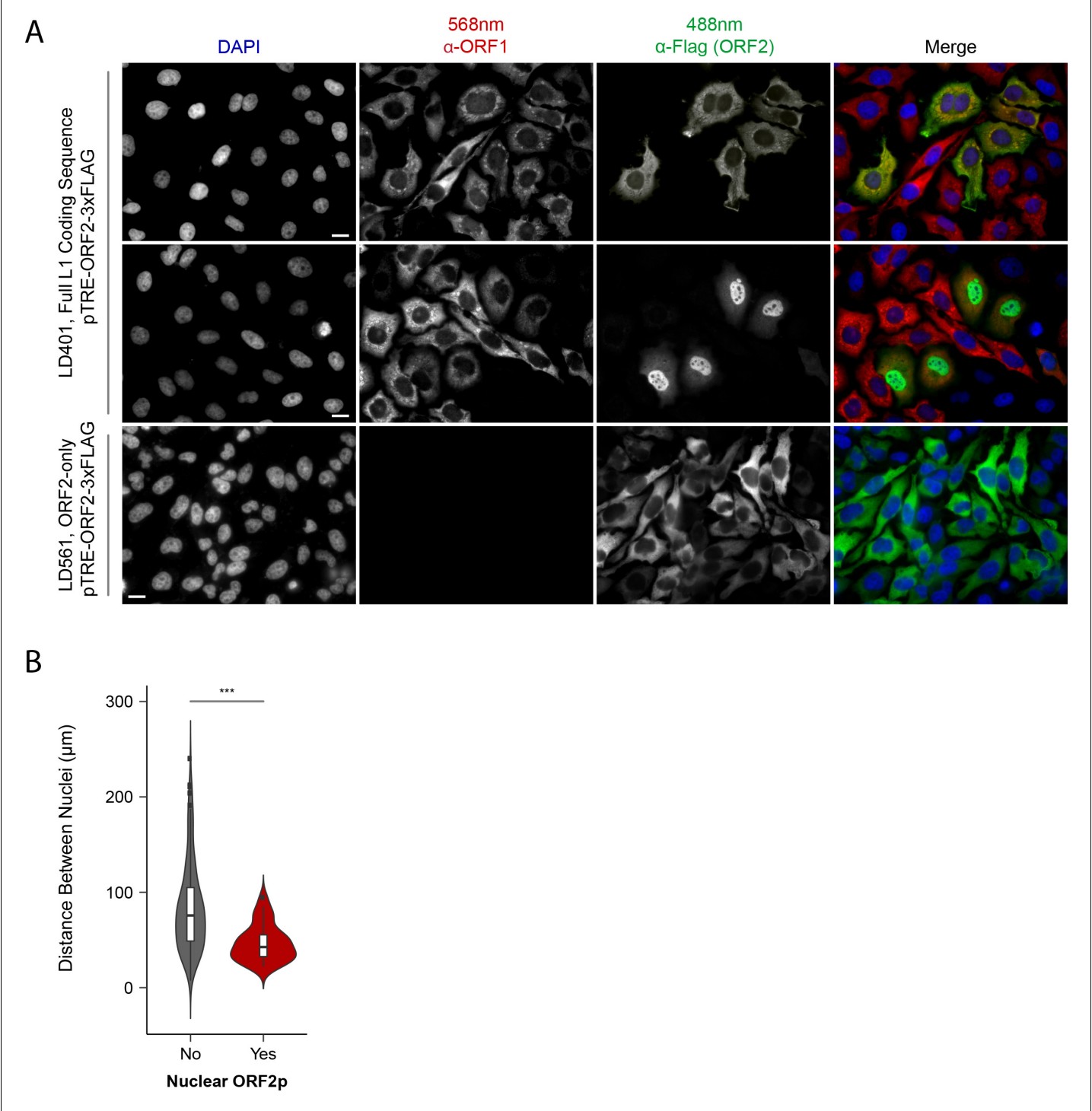

**Figure 3.** Immunofluorescent imaging reveals ORF1p expression is required for nuclear ORF2p staining. (**A**) Puromycin-selected HeLa-M2 cells containing pLD401 (Tet promoter, [*ORFeus*-Hs] full L1 coding sequence, ORF2p-3xFLAG, top two rows) or pLD561 (Tet promoter, ΔORF1, ORF2p-3xFLAG, bottom row) were plated on fibronectin-coated coverslips and induced for 24 hr with doxycycline prior to fixation and staining. With pLD401, the previously-observed pattern of cytoplasmic-only ORFs (top row) and a new pattern of pairs of cells displaying ORF2p in the nucleus (middle row) were apparent. When ORF1p was omitted from the construct (pLD561, bottom row), nuclear ORF2p was not apparent. Scale bars: 10 μm. (**B**) *Statistical analysis of the distances between pairs of ORF2p + nuclei as compared to random*: Violin plots of the distributions of shortest distances between 1000 pairs of randomly selected nuclei ('no') and the observed pairs of ORF2p + nuclei ('yes') in cells transfected with pLD401; n = 262 cells, 47 nuclear ORF2 +. ***p=$3.955 \times 10^{-11}$ (Welch's t-test).

*Figure 3 continued on next page*

*Figure 3 continued*

DOI: https://doi.org/10.7554/eLife.30094.006

The following source data is available for figure 3:

**Source data 1.** Source data used in the analysis of ORF2p+ inter nuclear distance analysis: *Figure 3* and *Supplementary file 5*.
DOI: https://doi.org/10.7554/eLife.30094.007

observed that both ORF proteins enter the nucleus of HeLa cells during mitosis, however, nuclear ORF1p does not seem to be physically associated with nuclear ORF2p (see Discussion). Taken together, the data obtained from the modified IF experiments aligned well with our proteomic and biochemical data; L1 expression resulted in *at least* two distinct populations: cytoplasmic complexes containing both ORF1p and ORF2p, and nuclear complexes containing ORF2p while potentially lacking ORF1p.

## The effects of retrotransposition-blocking point mutations on the interactomes of affinity captured L1 RNPs

Based on the hypothesis that our composite purifications contain bona fide nuclear intermediates, we decided to explore the effects of catalytic point mutations within the ORF2p endonuclease and reverse transcriptase domains, respectively. We reasoned that such mutants may bottleneck L1 intermediates at the catalytic steps associated with host gDNA cleavage and L1 cDNA synthesis, potentially revealing protein associations that are important for these discrete aspects of target-primed reverse transcription (TPRT), the presumed mechanism of L1 transposition (*Luan et al., 1993*; *Feng et al., 1996*; *Cost et al., 2002*). For this we used an H230A mutation to inactivate the endonuclease activity (EN⁻/pLD567), and a D702Y mutation to inactivate the reverse transcriptase activity (RT⁻/pLD624) (*Taylor et al., 2013*). *Figure 4* illustrates the approach and displays the findings of our assay. Broadly, while we observed comparable RNA-level properties between samples (*Figure 4B*, *Supplementary file 4*), our findings revealed several classes of distinctive protein-level behaviors (*Figure 4C*). Two classes of behavior appeared to be particularly striking: (1) the yield of constituents of cytoplasmic L1s was reduced, relative to WT, by the EN⁻ mutation, yet elevated by the RT⁻ mutation (*Figure 4C*, left side); and (2) numerous constituents of nuclear L1s were elevated in yield by the EN⁻ mutation but reduced or nominally unchanged, relative to WT, by the RT⁻ mutation (*Figure 4C*, right side). With respect to the second group, IPO7, NAP1L4, NAP1L1, FKBP4, HSP90AA1, and HSP90AB1 were all elevated in the EN⁻ mutants, potentially implicating these proteins as part of an L1 complex (or complexes) immediately preceding DNA cleavage. Notably, there is a third class of proteins, including PURA/B, PCNA, TOP1, and PARP1, that all respond similarly to both EN⁻ and RT⁻ mutants compared to WT, exhibiting reduced associations with the mutant L1s; although, the RT⁻ mutant showed a larger effect size on the PURA/B proteins. These data suggest that cleavage of the host genomic DNA by ORF2p fosters associations between L1 and this third class of proteins, but that interactions with PURA/B may be further enhanced by L1 cDNA production. Other nuclear L1 proteins: HSPA8, HAX1, HSPA1A, TUBB, and TUBB4B were increased in both mutants. To better visualize the range of behaviors exhibited by our proteins of interest, and the population at large, we cross-referenced the relative enrichments of each protein detected in both experiments, shown in *Figure 4D*. We noted the same striking trend mentioned above, that two seemingly opposite behavioral classes of interactors could also be observed globally among all proteins associating with ORF2p catalytic mutants (see *Figure 4C*, left side and right side, and *Figure 4D*), creating the crisscross pattern displayed (see also *Figure 4—figure supplement 1*). Notably, the pattern observed appears to track with the relative behavior of ORF1p, which, along with other cytoplasmic L1 factors is elevated in RT⁻ mutants and reduced in EN⁻ mutants. We therefore speculate that the sum of observed interactomic changes include effects attributable directly to the catalytic mutations as well as potential indirect effects resulting in increased cytoplasmic RNPs (including ORF1p) in the RT⁻ mutant.

## Dynamics of L1 RNPs in vitro

We next decided to measure the in vitro dynamics of proteins copurifying with affinity captured L1s, reasoning that proteins with comparable profiles are likely candidates to be physically linked to one

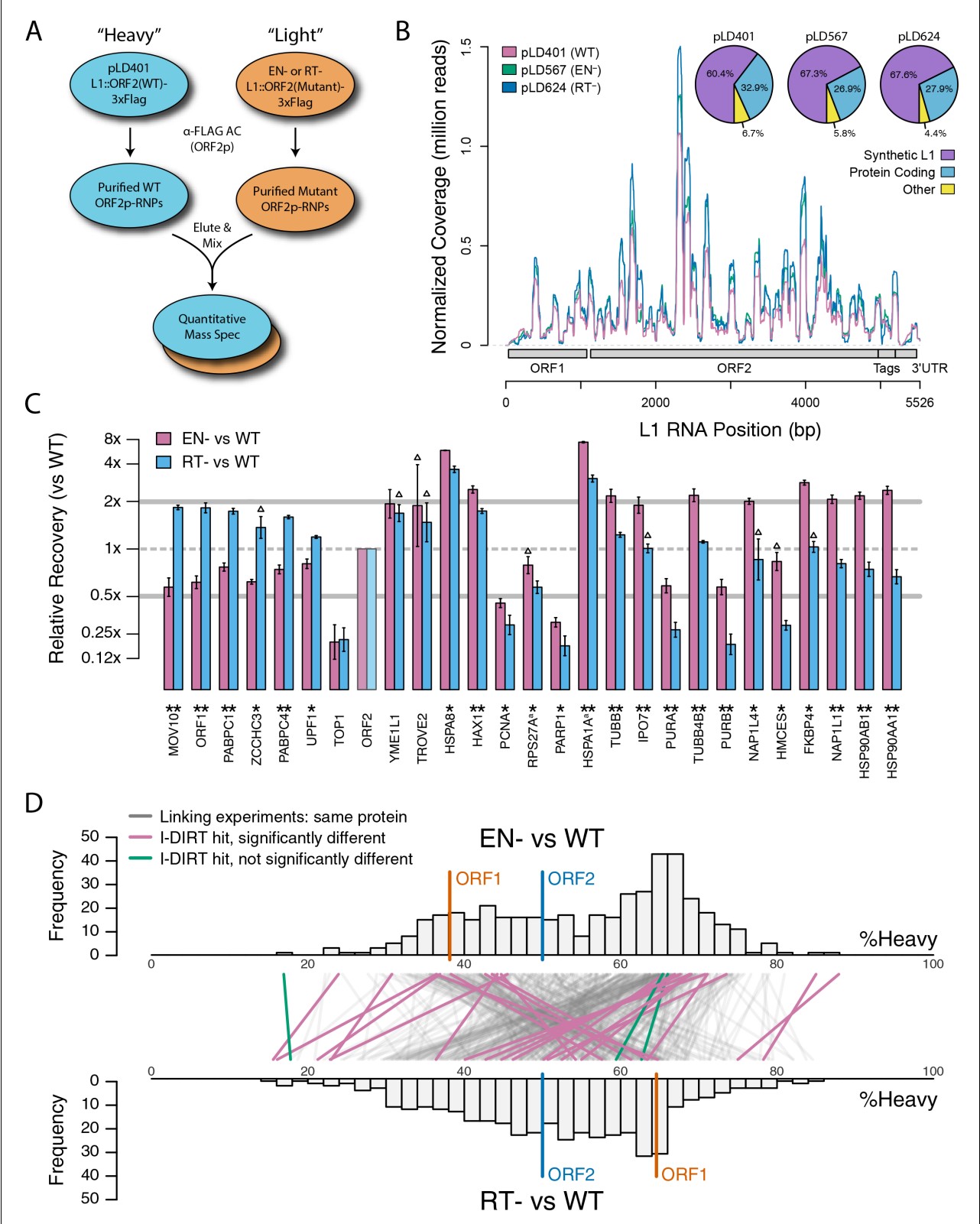

**Figure 4.** Catalytic inactivation of ORF2p alters the L1 interactome: L1s were affinity captured from cells expressing enzymatically active ORF2p-3xFLAG sequences (pLD401, WT), a catalytically inactivated endonuclease point mutant (pLD567; H230A, EN⁻), and a catalytically inactivated reverse transcriptase point mutant (pLD624; D702Y, RT⁻). These were analyzed by next-generation RNA sequencing and quantitative MS. (**A**) *Proteomic workflow*: WT L1s were captured from heavy-labeled cells, EN⁻ and RT⁻ L1s were captured from light-labeled cells. WT and either EN⁻ or RT⁻ fractions

*Figure 4 continued on next page*

*Figure 4 continued*

were mixed after affinity capture, in triplicate, and the relative abundance of each co-captured protein in the mixture was determined by quantitative MS. (B) *L1 RNA yield and coverage between different preparations*: As in *Figure 2A*, RNA extracted from 3xFLAG eluates originating from pLD401, pLD567, and pLD624 were subjected to next-generation sequencing. The results are summarized with respect to coverage of the synthetic L1 sequence (see schematic with nucleotide coordinates) as well as the relative quantities of mapped, annotated reads. The mean of duplicate experiments is displayed. (C) I-DIRT significant proteins displayed were detected in at least two replicates. All values were normalized to ORF2p. Data are represented as mean ±SD. Triangles (△) mark proteins whose levels of co-capture did not exhibit statistically significant differences in the mutant compared to the WT. A single or double asterisk denotes a statistically significant difference between the relative abundances of the indicated protein in EN⁻ and RT⁻ mutants: p-values of between 0.05–0.01 (*) and below 0.01 (**), respectively. Gray horizontal bars on the plot mark the 2x (upper) and 0.5x (lower) effect levels. (D) The double histogram plot displays the distributions of all proteins identified in at least two replicates, in common between both EN/WT (TOP) and RT/WT (LOWER) affinity capture experiments. The x-axis indicates the relative recovery of each copurifying protein and the y-axis indicates the number of proteins at that value (binned in two unit increments). The data are normalized to ORF2p. The relative positions of ORF2p and ORF1p are marked by colored bars. Differently colored lines illustrate the relative change in positions of the proteins within the two distributions (as indicated). Colored lines denote I-DIRT significance, with magenta lines indicating a statistically significant shift in position (p≤0.05) within the two distributions and green lines indicating that statistical significance was not reached (entities labeled in *Figure 4—figure supplement 1*). A cluster of magenta lines can be seen to track with ORF1p (red line, upper and lower histogram), and another cluster can be seen to behave oppositely, creating a crisscross pattern in the center of the diagram. A similar crisscross pattern is exhibited by many gray lines.

DOI: https://doi.org/10.7554/eLife.30094.008

The following figure supplement is available for figure 4:

**Figure supplement 1.** Double histogram plot with entities labeled.
DOI: https://doi.org/10.7554/eLife.30094.009

another or otherwise co-dependent for maintaining stable interactions with L1s. To achieve this, we first affinity captured heavy-labeled, affinity-tagged L1s and subsequently incubated them, while immobilized on the medium, with light-labeled, otherwise identically prepared cell extracts from cells expressing untagged L1s (*Luo et al., 2016*). In this scenario, heavy-labeled proteins present at the zero time point are effectively 'infinitely diluted' with light-labeled cell extract. The exchange of proteins, characterized by heavy-labeled proteins decaying from the immobilized L1s and being replaced by light-labeled proteins supplied by the cell extract, was monitored by quantitative MS. These experiments were conducted using constructs based on the naturally occurring L1RP sequence (*Dai et al., 2014*; *Taylor et al., 2013*; *Kimberland et al., 1999*). *Figure 5* illustrates the approach and displays the findings of our assay. We observed three distinctive clusters of behaviors (*Figure 5B,C*). Notably, ORF1p, ZCCHC3, and the cytoplasmic poly(A) binding proteins clustered together, forming a relatively stable core complex. Exhibiting an intermediate level of relative in vitro dynamics, UPF1 and MOV10 clustered with TUBB, TUBB4B, and HSP90AA1. A third, and least stable, cluster consisted of only nuclear L1 interactors.

## Multidataset integration

Having observed coordinated and distinctive behaviors exhibited by groups of L1 interacting proteins across several distinctive biochemical assays, we then integrated the data and calculated the behavioral similarity of the I-DIRT-significant interactors, producing a dendrogram; *Figure 6* displays their relative similarities. A cluster containing the putative cytoplasmic L1 components (MOV10, UPF1, ZCCHC3, PABC1/4, ORF1p) was observed, as was a cluster containing PURA/B, PCNA, TOP1, PARP1, aligning with our assessments of the separated datasets (*Figures 1*, *4* and *5*). In addition to these, we also observed three distinctive clusters derived from the nuclear L1 interactome. We believe that this is likely to reflect the presence of a collection of distinctive macromolecules.

## Discussion

In this study we have characterized biochemical, interatomic, enzymatic, and cellular localization properties of ectopically expressed L1s. Through the assays explored, we observed discrete and coordinated behaviors, permitting us to refine our model of L1 intermediates, diagrammed in *Figure 7*. We propose a cytoplasmic L1, composed of ORF1/2 p, L1 RNA, PABPC1/4, MOV10, UPF1, and ZCCHC3, that constitutes an abundant, canonical RNP intermediate often referred to in the literature. MOV10, UPF1, and ZCCHC3 are apparently substoichiometric to ORF2p in our preparations, therefore it may be that only a subset of cytoplasmic intermediates engages these host

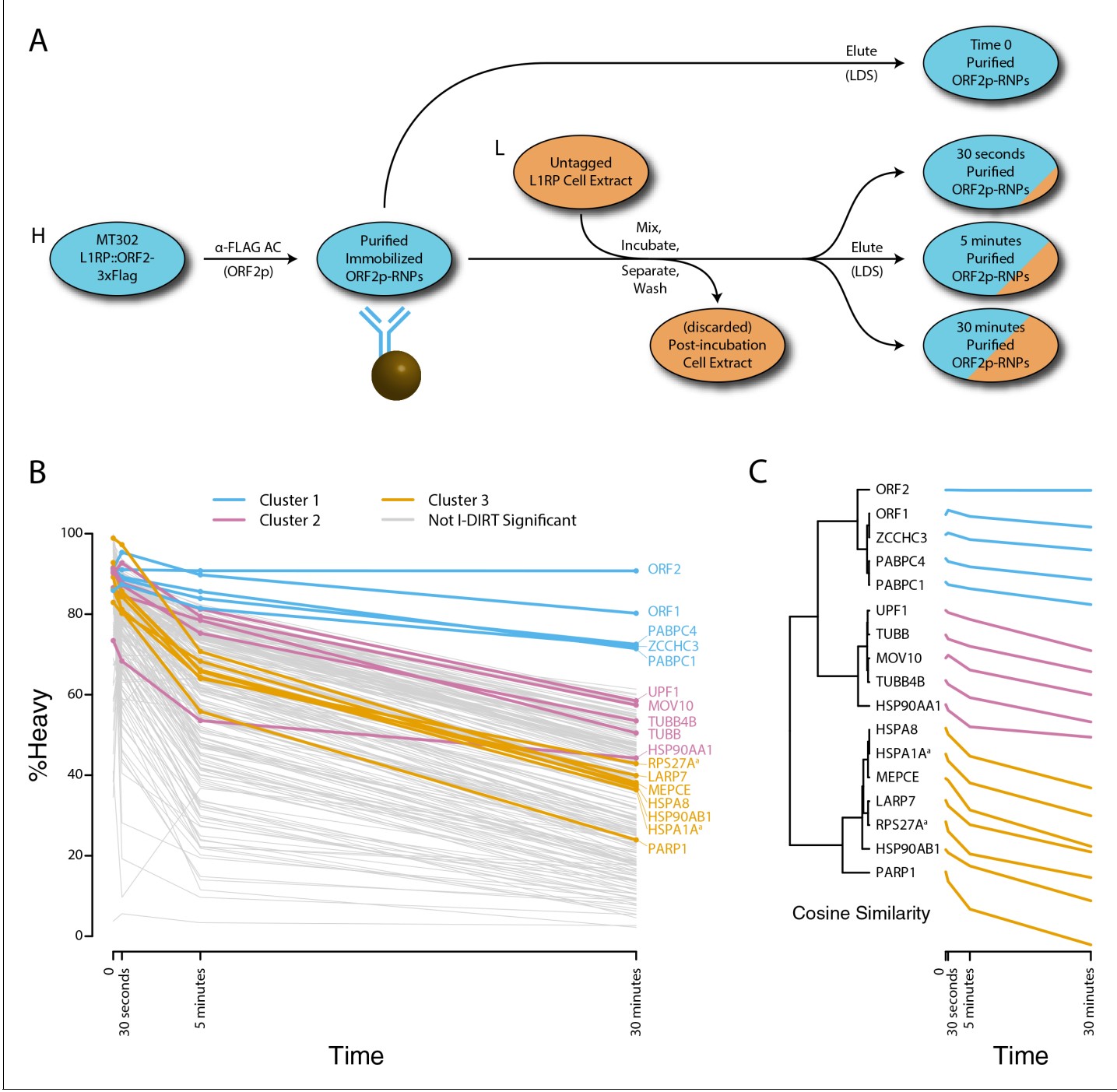

**Figure 5.** Monitoring coordinated dissociation and exchange exhibited by L1 interactors in vitro: L1s were affinity captured from heavy-labeled cells expressing ORF2p-3xFLAG in the context of the naturally occurring L1RP sequence (pMT302); the stabilities of the protein constituents of the captured heavy-labeled L1 population were monitored in vitro by competitive exchange with light-labeled cell extracts containing untagged L1s (pMT298) (***Taylor et al., 2013***). (**A**) 3xFLAG-tagged L1s were captured from heavy-labeled cells and then, while immobilized on the affinity medium, were treated with an otherwise identically prepared, light-labeled, untagged-L1-expressing cell extract. Untreated complexes were compared to independently prepared complexes incubated for 30 s, 5 min, and 30 min, (respectively) to determine the relative levels of in vivo assembled heavy-labeled interactors and in vitro exchanged light-labeled interactors, using quantitative MS. (**B**) The results were plotted to compare the percentage of heavy-labeled protein versus time. I-DIRT significant proteins from ***Table 1*** are highlighted if present. Three clusters were observed (as indicated). (**C**) The cosine distance between the observed I-DIRT significant proteins was plotted along with time.

DOI: https://doi.org/10.7554/eLife.30094.010

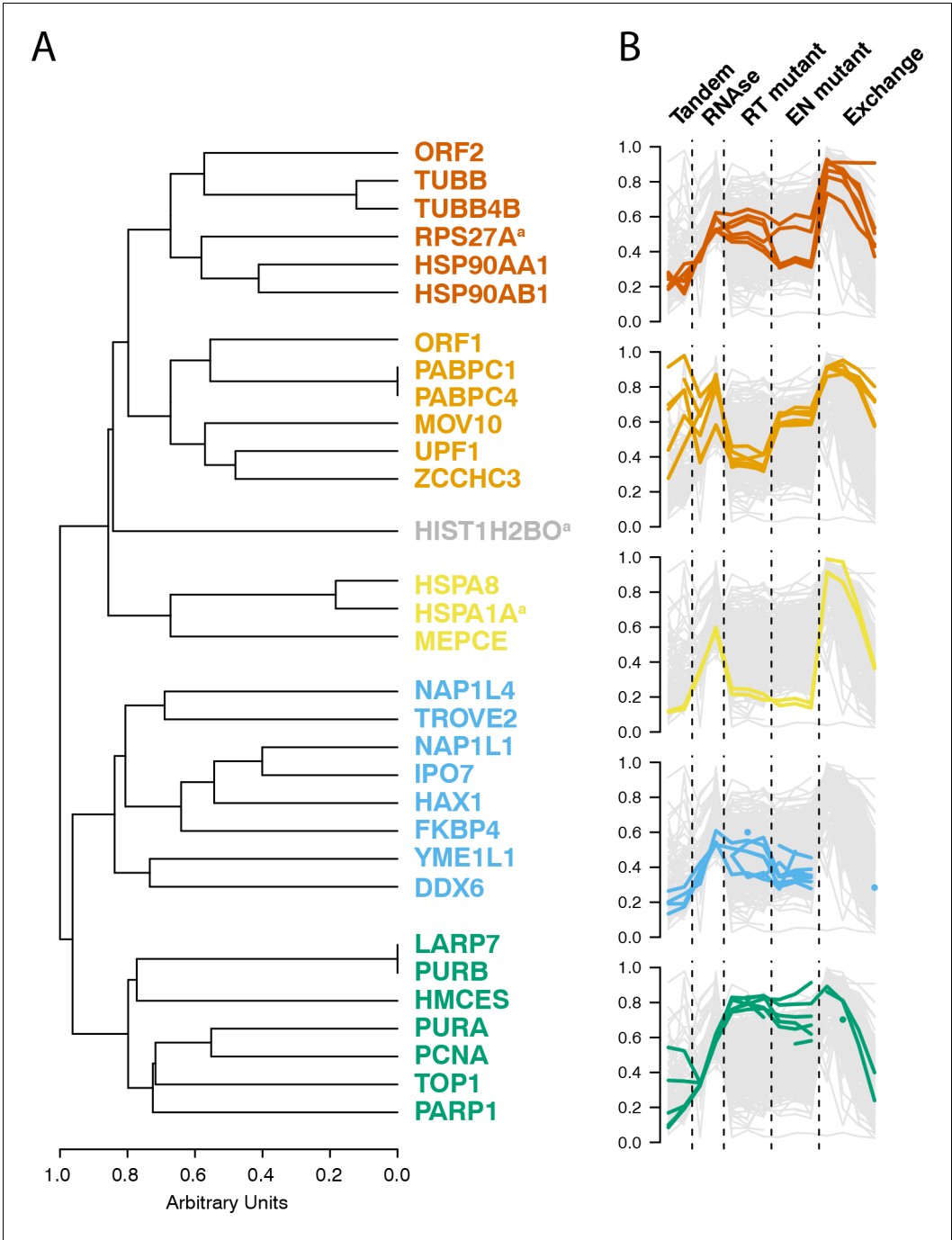

**Figure 6.** Interactomic data integration  (A) All MS-based affinity proteomic experiments presented were combined and analyzed for similarities across all I-DIRT significant proteins, producing five groupings. Distance are presented on a one-unit arbitrary scale (see Materials and methods: Mass Spectrometry Data Analysis). (B) The traces of each protein in each cluster, across all experiments, are displayed. The y-axis indicates the raw relative-enrichment value and the x-axis indicates the categories of each experiment-type. Each category is as wide as the number of replicates or time-point samples collected.

DOI: https://doi.org/10.7554/eLife.30094.011

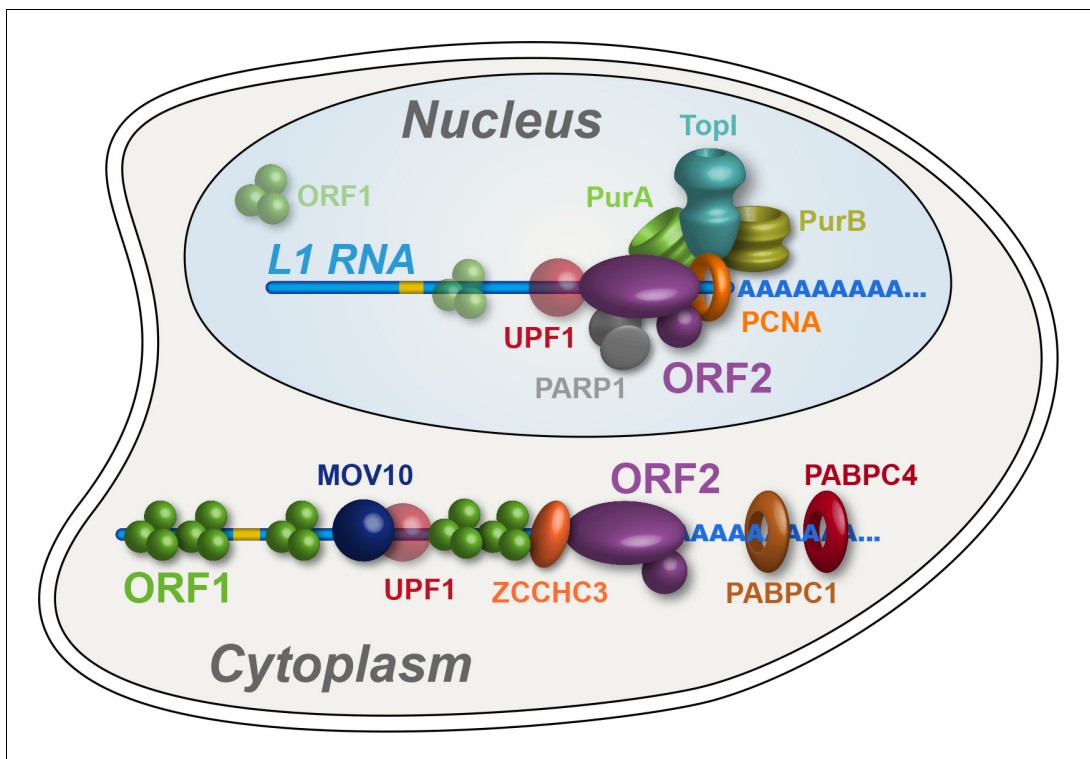

**Figure 7.** Refined interatomic model: Our results support the existence of distinct cytoplasmic and nuclear L1 interactomes. Affinity capture of L1 via 3xFLAG-tagged ORF2p from cell extracts results in a composite purification consisting of several macromolecular (sub)complexes. Among these, we propose a canonical cytoplasmic L1 RNP (**depicted**) and one or more nuclear macromolecules. UPF1 exhibited equivocal behavior within our fractionations and was also co-captured with chromatin associated ORF2p, suggesting it participates in both cytoplasmic and nuclear L1 interactomes. Within the nuclear L1 interactome, our data support the existence of a physically linked entity consisting of (at least) PCNA, PURA/B, TOP1, and PARP1 (**depicted**).

DOI: https://doi.org/10.7554/eLife.30094.012

restriction factors. On the other hand, this apparent relative abundance may simply reflect a lower in vitro stability of UPF1 and MOV10 within this complex (*Figure 5*). Although ORF1p is apparently required for efficient ORF2p nuclear entry, we also propose a second more complicated population, lacking (or with significantly less) ORF1p, that constitutes nuclear or pre- dominantly nuclear L1 macromolecules. We note that *Alu* elements exhibit ORF2p-dependent mobilization that does not require ORF1p, but appears to be enhanced by ORF1p in some contexts (*Dewannieux et al., 2003*; *Wallace et al., 2008*); this is not true for L1 or processed pseudogenes, and we conclude *Alu* RNPs likely exploit an alternate mechanism of nuclear entry. The nuclear L1 population is enriched for factors linked to DNA replication and repair, including PURA, PURB, PCNA, TOP1, and PARP1; we propose that these proteins, along with ORF2p, form part of a direct intermediate of TPRT, although these components may not all act in synergy. Our proposals are broadly supported by the findings of *Mita et al., 2018*, who present data to support the hypothesis that PCNA-associated ORF2p is not appreciably associated with ORF1p, and also identified TOP1 and PARP1 in complex with ORF2p/PCNA.

Although the protein purification approach was the similar, we observed an apparently larger proportion of L1 RNA in our recent preparations than in our previous study. We reported that L1 constituted ~25% of mapped reads previously (*Taylor et al., 2013*); a comparable result was obtained when we reanalyzed that data using the pipeline described here (see Materials and methods): ~93% of reads in our reanalyzed 2013 dataset mapped to the human genome, and L1 constituted ~20% of reads mapped to annotated features ('annotated reads') in 3xFLAG eluates. In this study we report that ~60% of annotated reads mapped to synthetic L1 in 3xFLAG eluates (*Figure 2A*). The higher proportion of L1 recovered may be due to the combination of higher fidelity

RNA preparative methods and advanced sequencing technology used here; we observed ~10x more total reads mapping to L1 and comparatively improved, more uniform coverage across the entire L1 sequence, likely explaining the discrepancy. We also noted that the number of normalized reads mapped to L1 in our initial 3xFLAG elutions ('input') and subsequent tandem-purified α-ORF1p elutions were comparable, and yet ~1/2 as many were seen in the α-ORF1p supernatant fraction (*Figure 2A,B*). We suspect that this is due to saturation in library preparations or sequencing steps for the 'input' and 'elution' fractions, but conclude that more L1 RNA is in the 'cytoplasmic' elution fraction than the 'nuclear' supernatant.

We observed substantial and comparable LEAP activity in both our tandem-purified ORF1p+ ('elution') and ORF1p– (immuno-depleted 'supernatant') populations (*Figure 2C*, *Supplementary file 4*). To our knowledge, these represent the simplest and purest endogenously assembled L1 RNPs yet reported that exhibit robust signal in the LEAP assay. We note that, our results demonstrating robust activity in the nuclear-enriched supernatant fraction (depleted of ORF1p) may contrast with previous reports of reduced LEAP activity in constructs where ORF1p RNA binding was compromised (*Kulpa and Moran, 2006*), but our fractions merit further study and comparisons on the basis of ORF2p and RNA levels to determine specific activity.

## Cytoplasmic L1 macromolecules

ORF1p, MOV10, UPF1, and ZCCHC3 are released from L1 RNPs by treatment with RNases (*Figure 1*), indicating the importance of the L1 RNA in the maintenance of these interactions. In this context, the L1 ORF and poly(A) binding proteins support L1 proliferation (*Kulpa and Moran, 2006*; *Dai et al., 2012*; *Wei et al., 2001*), whereas ZCCHC3 (*Supplementary file 2*) and MOV10 (*Goodier et al., 2012*; *Arjan-Odedra et al., 2012*) function in repressive capacities. Although UPF1 might also be expected to operate in a repressive capacity through its role in nonsense mediated decay (NMD), we previously demonstrated that UPF1's role does not apparently resemble that of canonical NMD and it acts as an enhancer of retrotransposition despite negatively affecting L1 RNA and protein levels, supporting the possibility of repressive activity in the cytoplasm and proliferative activity in the nucleus (*Taylor et al., 2013*). Notably, MOV10 has been implicated in the recruitment of UPF1 to mRNA targets through protein-protein interactions (*Gregersen et al., 2014*). However, we observed that MOV10 exhibited a greater degree of RNase-sensitivity than UPF1, indicating that, if MOV10 directly modulates the UPF1 interactions with L1, a sub-fraction of UPF1 exhibits a distinct behavior (UPF1 is ~62% as sensitive to RNase treatment as MOV10, *Figure 1C*). Bimodal UPF1 behavior can also be seen in split-tandem capture experiments, only about half of the UPF1 exhibited ORF1p-like partitioning with the canonical L1 RNP (Figure 1F). Moreover, UPF1 was recovered with L1s affinity captured from fractionated chromatin (further discussed below), and only about half of the UPF1 exhibits ORF1p-like partitioning with the canonical L1 RNP (*Figure 1F*). Presumably, the RNase-sensitive fraction, released in concert with MOV10, is the same fraction observed in cytoplasmic L1s obtained by split-tandem capture. In contrast, PABPC1 and C4 exhibit strong ORF1p-like partitioning (comparable to MOV10), but appear wholly insensitive to RNase treatment. This is most likely due to the fact that neither RNase A nor T1 cleave RNA at adenosine residues (*Volkin and Cohn, 1953*; *Yoshida, 2001*); hence poly(A) binding proteins may not be ready targets for release from direct RNA binding by the assay implemented here (or generally, using these ribonucleases). Failure to release ORF2p into the supernatant upon RNase treatment is expected due to its immobilization upon the affinity medium (*Dai et al., 2014*). However, we note that ORF2p binding to the L1 RNA has also been proposed to occur at the poly(A) tail (*Doucet et al., 2015*), raising the related possibility of close physical association on the L1 RNA between ORF2p and PABPC1/4 in cytoplasmic L1 RNPs. ORF1p, PABPC1/4, MOV10, ZCCHC3, and UPF1, all behaved comparably in response to EN⁻ and RT⁻ catalytic mutations, decreasing together in EN⁻ mutants, and increasing together in RT⁻ mutants (*Figure 4C*). Moreover, when the exchange of proteins within L1 RNPs was monitored directly, PABPC1/4 and ZCCHC3 exhibited nearly identical stability, well above the background distribution; UPF1 and MOV10 also exhibited comparable kinetics to one another, falling into an intermediary stability cluster (*Figure 5B,C*).

RNase-sensitivity was displayed by numerous proteins not previously identified as putative L1 interactors (*Table 1*, *Figure 1*; [*Taylor et al., 2013*]). A known limitation of I-DIRT (and many SILAC-based analyses) is that it cannot discriminate non-specific interactors from specific but rapidly exchanging interactors (*Wang and Huang, 2008*; *Luo et al., 2016*; *Smart et al., 2009*). Our samples

likely contain rapidly exchanging, physiologically relevant factors that were not revealed by I-DIRT under the experimental conditions used. With this in mind, we note members of the exon junction complex (EJC), RBM8A (Y14), EIF4A3 (DDX48), and MAGOH, are among our RNase-sensitive constituents, all exhibiting a similar degree of RNase-sensitivity (*Figure 1C*, labeled black dots). Crucially, these proteins are physically and functionally connected to UPF1 (reviewed in [*Schweingruber et al., 2013*]), and physically to MOV10 (*Gregersen et al., 2014*), both validated L1 interactors. We therefore hypothesize that EJCs may constitute bona fide L1 interactors missed in our original screen. This may seem unexpected because canonical L1 RNAs are thought not to be spliced, but this assumption has been challenged by one group (*Belancio et al., 2006*), and splicing-independent recruitment of EJCs has also been demonstrated (*Budiman et al., 2009*). Perhaps more compelling, EJC proteins exhibited a striking similarity in RNase-sensitivity to MOV10 (*Figure 1C*). EIF4A3 has been suggested to form an RNA-independent interaction with MOV10 (*Gregersen et al., 2014*), and MOV10 is a known negative regulator of L1, making it attractive to speculate that these proteins were recruited and released in concert with MOV10 and/or UPF1.

Ectopically expressed canonical L1 RNPs have been shown to accumulate in cytoplasmic stress granules (*Doucet et al., 2010*; *Goodier et al., 2010*), and our observation of UPF1, MOV10, and MAGOH in the RNase-sensitive fraction is consistent with this characterization (*Jain et al., 2016*). However, the additional presence of EIF4A3 and RBM8A suggested that our RNPs may instead overlap with IGF2BP1 (IMP1) granules, reported to be distinct from stress granules (*Jønson et al., 2007*; *Weidensdorfer et al., 2009*). Consistent with this possibility, we observed IGF2BP1, YBX1, DHX9, and HNRNPU within the mixture of co-captured proteins (*Supplementary file 1*). We did not, however, observe canonical stress granule markers G3BP1 or TIA1 (*Goodier et al., 2007*; *Jain et al., 2016*; *Doucet et al., 2010*). Surprisingly, siRNA knockdown of IGF2BP1 substantially reduced L1 retrotransposition; however, we note that the cytotoxicity associated with knocking-down EJC components may confound interpretation (*Supplementary file 2*). Given the result obtained, IGF2BP1 appears to support L1 proliferation. Consistent with an established function (*Bley et al., 2015*; *Weidensdorfer et al., 2009*), IGF2BP1 granules may sequester and stabilize L1 RNPs in the cytoplasm, creating a balance of L1 supply and demand that favors proliferation over degradation. Although human L1 does not contain a known IRES, it is known that *ORF2* is translated by a noncanonical mechanism (*Alisch et al., 2006*), and IGF2BP1 may promote this (*Weinlich et al., 2009*).

## Nuclear L1 macromolecules

The fraction eluted from the α-ORF1p medium contained the population of proteins physically linked to both ORF2p and ORF1p and greatly resembled the components released upon RNase treatment, hence these linkages primarily occur through the L1 RNA (or are greatly influenced by it). In contrast, the supernatant from the α-ORF1p affinity capture contained the proteins we speculate to be associated with ORF2p, but not ORF1p; moreover, fully intact RNA does not appear to be essential to the maintenance of these interactions. An exciting alternate interpretation to direct protein-protein linkage is that some of the L1 RNAs in this population may be at least partially hybridized to L1 cDNAs, which would render them RNase resistant: at the salt concentration used in our RNase assay (0.5 M; *Figure 1C*), RNase A is unlikely to cleave the RNA component of DNA/RNA hybrids (*Halász et al., 2017*; *Wyers et al., 1973*), and such activity is not expected of RNase T1. This interpretation is supported by several pieces of indirect evidence: (1) the presence of well-known DNA binding factors (*Figure 1*); (2) the presence of several of these same factors (PARP1, PCNA, PURA, and TOP1) in ORF2p-3xFLAG affinity captured from enriched chromatin (*Supplementary file 3*); (3) The pronounced decrease in stable in vivo co-assembly of TOP1, PCNA, PARP1, PURA, and PURB in affinity captured L1 fractions harboring ORF2p EN$^-$ and RT$^-$ mutations (*Figure 4*), with a greater effect in RT$^-$ mutations; and (4) our L1 preparations exhibit RT activity (*Figure 2C*, in vitro; as well as in vivo [*Taylor et al., 2013*]). If true, linkage of subcomplexes via DNA/RNA hybrids would further support the nuclear origin of much of this fraction; further study is needed. Notable within this group of putative nuclear interactors was the PURA/PURB/PCNA cluster (*Figure 1F*), with TOP1 also in close proximity, ontologically grouping to the nuclear replication fork (GO:0043596). Separately, a few physical and functional connections have been shown for PURA/PURB (*Knapp et al., 2006*; *Kelm et al., 1999*; *Mittler et al., 2009*), PCNA/TOP1 (*Takasaki et al., 2001*), and PURA/PCNA (*Qin et al., 2013*). Notably, PURA, PURB, and PCNA have been independently linked to replication-factor-C/replication factor-C-like clamp loaders (*Kubota et al., 2013*; *Havugimana et al., 2012*). Given that

we also observe tight clustering of protein pairs known to be physically and functionally linked, e.g. PABPC1/4 (*Jønson et al., 2007*; *Katzenellenbogen et al., 2007*) and HSPA8/1A (*Jønson et al., 2007*; *Nellist et al., 2005*), and because we have established PCNA as a positive regulator of L1 retrotransposition (*Taylor et al., 2013*), we propose that the [PURA/B/PCNA/TOP1] group is a functional sub-complex of nuclear L1. In addition, although it does not cluster as closely to the [PURA/B/PCNA/TOP1] group, PARP1 is found within the putative nuclear L1 population and is functionally linked with PCNA, specifically stalled replication forks (*Bryant et al., 2009*; *Min et al., 2013*; *Ying et al., 2016*). Further tying them together, these proteins all also exhibited substantial affinity capture yield decreases in response to mutations that abrogated ORF2p EN or RT activity (*Figure 4*). This is compelling because these ORF2p enzymatic activities are required in order for it to manipulate DNA and traverse the steps of the L1 lifecycle that benefit from physical association with replication forks. One caveat to this interpretation is that, while knocking down PCNA reduced L1 retrotransposition (*Taylor et al., 2013*), no such effect was observed for TOP1 or PURA/B, which led instead to mild increases in L1 activity (*Supplementary file 2*). These proteins may be physically assembled within a common intermediate, but functionally antagonistic. HSP90 proteins were also observed in this fraction, and are also linked with stalled replication forks (*Arlander et al., 2003*; *Ha et al., 2011*), but exhibited a distinctive response to catalytic mutants, accumulating in $EN^-$ mutants while exhibiting a modest decrease in $RT^-$ mutants. The recruitment of the ORF2p/PCNA complex to stalled replication forks has been also proposed by *Mita et al., 2018*.

As mentioned above, we previously speculated that an RNase-insensitive fraction of L1-associated UPF1 may support retrotransposition in conjunction with PCNA in the nucleus (*Azzalin and Lingner, 2006*; *Taylor et al., 2013* and *Mita et al., 2018*). In contrast to other PCNA-linked proteins, catalytic inactivation of ORF2p did not robustly affect the relative levels of co-captured UPF1, and UPF1 behaved in a distinct manner during tandem capture. The equivocal behavior of UPF1 in several assays (*Figures 1*, *4* and *5*) supports UPF1's association with both the putative cytoplasmic and nuclear L1 populations, the latter being additionally supported by the association of UPF1 with ORF2p-3xFLAG captured from chromatin (*Supplementary file 3*). NAP1L4, NAP1L1, FKBP4, HSP90AA1, and HSP90AB1 (*Baltz et al., 2012*; *Castello et al., 2012*; *Simon et al., 1994*; *Rodriguez et al., 1997*; *Peattie et al., 1992*) are associated with RNA binding, involved in protein folding and unfolding, and function as nucleosome chaperones. An interesting possibility is that they have a nucleosome remodeling activity that may be required to allow reverse transcription to begin elongating efficiently, or for assembly of nucleosomes on newly synthesized DNA.

## Future studies

An obvious need is the continued validation of putative interactors by in vivo assays. Genetic knock-downs coupled with L1 insertion measurements by GFP fluorescence (*Ostertag et al., 2000*) provide a powerful method to detect effects on L1 exerted by host factors. However, this approach can sometimes be limited by cell viability problems associated with important genes; it is therefore critical to control for this (*Supplementary file 2*). IF and high-resolution microscopy may be useful to demonstrate co-localization of putative L1-associated proteins and may also be informative, warranting effort to identify appropriate antibodies and assay conditions. Bolstered by our analytical successes, RNA-sequencing, LEAP, and RNase-based affinity proteomics appear as notably high-value assays for further application-specific expansion and refinement.

Throughout this and our prior study (*Taylor et al., 2013*) we have used comparable in vitro conditions for the capture and analysis of L1 interactomes. However, we are aware that this practice has enforced a single biochemical 'keyhole' through which we have viewed L1-host protein associations. It is important to expand the condition space in which we practice L1 interactome capture and analysis in order to expand our vantage point on the breadth of L1-related macromolecules (*Hakhverdyan et al., 2015*). In concert with this, we must develop sophisticated, automated, reliable, low-noise methods to integrate biochemical, proteomic, genomic, and ontological data; the first stages of which we have attempted in the present study. Although we have used I-DIRT to increase our chances of identifying bona fide interactors (*Tackett et al., 2005*; *Taylor et al., 2013*), it is clear, and generally understood, that some proteins not making the significance cut-off will nevertheless prove to be critical to L1 activity (*Byrum et al., 2011*; *Luo et al., 2016*; *Joshi et al., 2013*), such as demonstrated by our unexpected findings with IGF2BP1 (*Supplementary file 2*). Through further development, including reliable integration with diverse, publicly available interactome

studies, we hope to enable the detection of extremely subtle physical and functional distinctions between (sub)complexes and their components, considerably enhancing reliable exploration and hypothesis formation. Furthermore, it is striking that no structures of assembled L1s yet exist; these are missing data that are likely to provide a profound advance for the mechanistic understanding of L1 molecular physiology. However, we believe that with the methods presented here, endogenously assembled ORF1p/ORF2p/L1-RNA-containing cytoplasmic L1 RNPs can be prepared at sufficiently high purity and yield (*Figure 1F*) to enable electron microscopy studies. Importantly, we have shown that our affinity captured fractions are enzymatically active for reverse transcription of the L1 RNA (*Figure 2C*; (*Taylor et al., 2013*)), providing some hope that cryo-electron microscopy could be used to survey the dynamic structural conformations of L1s formed during its various lifecycle stages (*Takizawa et al., 2017*).

# Materials and methods

## Key resources table

| Reagent type (species) or resource | Designation | Source or reference | Identifiers | Additional information |
|---|---|---|---|---|
| gene (human) LINE-1 | *ORFeus*-Hs; L1RP | 10.1016/j.cell.2013.10.021; 10.1186/1759-8753-2-2 | | |
| cell line (human) | HEK-293T_LD | 10.1016/j.cell.2013.10.021; 10.1128/MCB.06785–11 | | Mycoplasma testing was done regularly and was negative. We received an authenticated cell line from the ATCC and subsequently made them blastomycin resistant so we validated cells by blastomycin resistance. |
| transfected construct (human) | pLD401; pLD561; pLD567; pLD624; pMT302; pMT289 | 10.1016/j.cell.2013.10.021 | | |
| antibody | anti-FLAG; anti-ORF1p | 10.1016/j.cell.2013.10.021 | Sigma-Aldrich Cat# F1804, RRID:AB_262044; custom made, Abmart: 4H1 | |
| peptide, recombinant protein | ORF1p N-terminal di-peptide | this paper | | |
| software, algorithm | Scripts for IF (*Figure 3*); formal analysis used custom R code throughout | this paper | | Scripts are in *Supplementary file 5*; R code in – https://bitbucket.org/altukhov/line-1/ |

The preparation of L1 RNPs was carried out essentially as previously described (*Taylor et al., 2013*, *Taylor et al., 2016*), with modifications described here. Briefly, HEK-293T$_{LD}$ cells (*Dai et al., 2012*) transfected with L1 expression vectors were cultured as previously described or using a modified suspension-growth SILAC strategy described below. L1 expression was induced with with 1 µg/ml doxycycline for 24 hr, and the cells were harvested and extruded into liquid nitrogen. In all cases the cells were then cryogenically milled (*LaCava et al., 2016*) and used in affinity capture experiments and downstream assays. Custom computer code written in the R programming language was used in the analysis of mass spectrometry and RNA sequencing data; it has been published on https://bitbucket.org (*Altukhov, 2017*); a copy is archived at https://github.com/elifesciences-publications/altukhov-line-1.

## Modified SILAC strategy

Freestyle-293 medium lacking Arginine and Lysine was custom-ordered from Life Technologies, and heavy or light amino acids plus proline were added at the same concentrations previously described (*Taylor et al., 2013*), without antibiotics. Suspension-adapted HEK-293T$_{LD}$ were spun down, transferred to SILAC medium and grown for >7 cell divisions in heavy or light medium. On day 0, four (4) 1L square glass bottles each containing 200 ml of SILAC suspension culture at ~2.5 × $10^6$ cells/ml were transfected using 1 µg/ml DNA and 3 µg/ml polyethyleneimine 'Max' 40 kDa (Polysciences, Warrington, PA, #24765). A common transfection mixture was made by pre-mixing 800 µg DNA and

2.4 mL of 1 mg/ml PEI-Max in 40 ml Hybridoma SFM medium (Life Technologies, Grand Island, NY, #12045–076) and incubating for 20 min at room temperature (RT); 10 ml of the mixture was added to each bottle. On day 1, cells (200 ml) were split 1:2.5 (final two bottles each containing 250 mL) without changing the medium. On day 3, the cells were induced with 1 µg/ml doxycycline, and on day four the cells were harvested and extruded into liquid nitrogen. Aliquots were tested by western blot and the per-cell expression of both ORFs was indistinguishable from puromycin-selected material described previously (Appendix 1); transfection efficiency was assessed at >95% by indirect immunofluorescence of expressed ORF proteins. The median lysine and arginine heavy isotope incorporation levels for cell lines presented in this study were >90%, determined as previously described (*Taylor et al., 2013*).

## RNase-sensitivity affinity capture

Four sets of 200 mg of light (L) and heavy (H) pLD401 transfected cell powders, respectively, were extracted 1:4 (w:v) with 20 mM HEPES-Na pH 7.4, 500 mM NaCl, 1% (v/v) Triton X-100 (extraction solution), supplemented with 1x protease inhibitors (Roche, Indianapolis, IN, #11836170001). After centrifugal clarification, all of the L and H supernatants were pooled, respectively, and then split, resulting in two sets of cleared L and H extracts equivalent to duplicate 400 mg samples from each SILAC cell powder. These four samples were each subjected to affinity capture upon 20 µl α-FLAG magnetic medium. After binding and washing, one set of L and H samples were treated with a control solution consisting of 2 µl of 2 mg/ml BSA (Thermo Fisher Scientific, Waltham, MA, #23209) and 50 µl extraction solution, v:v (Ctrl); the other set of L and H samples was treated with a solution of 2 µl 2 mg/ml RNase A/5000 u/ml RNase T1 (Thermo Fisher Scientific #EN0551) and 50 µl extraction solution, v:v (RNase). Samples were then incubated 30 min at RT with agitation, the supernatant was removed, and the medium was washed three times with 1 ml of extraction solution. The retained captured material was eluted from the medium by incubation with 40 µl 1.1x LDS sample loading buffer (Life Technologies #NP0007). To enable quantitative comparisons of fractions, the samples were combined, respectively, as follows: 30 ul each of the $^L$RNase with $^H$Ctrl, and 30 ul each of the $^L$Ctrl with $^H$RNase. These samples were reduced, alkylated and run until the dye front progressed ~6 mm on a 4–12% Bis-Tris NuPAGE gel (Life Technologies, as per manufacturer's instructions). The gels were subsequently subjected to colloidal Coomassie blue staining (*Candiano et al., 2004*) and the sample regions ('gel-plugs') excised and processed for MS analyses, as described below.

## Split-tandem affinity capture

400 mg of light (L) and heavy (H) pLD401 transfected cell powders, respectively, were extracted and clarified as above. These extracts were subjected to affinity capture on 20 µl α-FLAG magnetic medium, 30 min at 4°C, followed by native elution with 50 µl 1 mg/ml 3xFLAG peptide (15 min, RT). 45 µl of the elution were subjected to subsequent affinity capture upon 20 µl α-ORF1 magnetic medium, resulting in a 45 µl supernatant (Sup) fraction depleted of ORF1p. Finally, the material was eluted (Elu) from the α-ORF1p medium in 45 µl 2.2x LDS sample loading buffer by heating at 70°C for 5 min with agitation. To enable quantitative comparisons of fractions the samples were combined, respectively, as follows: 28 µl each of the $^L$Sup with $^H$Elu, and 28 µl each of the $^L$Elu with $^H$Sup. These samples were then prepared as gel-plugs (as above) and processed for MS analyses, as described below.

## Mass spectrometry sample preparation and data acquisition

Gel plugs were excised, cut into 1 mm cubes, de-stained, and digested overnight with enough 3.1 ng/µl trypsin (Promega, Madison, WI, #V5280) in 25 mM ammonium bicarbonate to cover the pieces. In RNase-sensitivity and split-tandem SILAC analyses based on pLD401, as well as in vitro protein exchange experiments based on pMT302 and pMT289, an equal volume of 2.5 mg/ml POROS R2 20 µm beads (Life Technologies #1112906) in 5% v/v formic acid, 0.2% v/v TFA was added, and the mixture incubated on a shaker at 4°C for 24 hr. Digests were desalted on Stage Tips (*Rappsilber et al., 2007*), eluted, and concentrated by vacuum centrifuge to ~10 µl. ~3 µl were injected per LC-MS/MS analysis. RNase-sensitivity and split-tandem samples were loaded onto a PicoFrit column (New Objective, Woburn, MA) packed in-house with 6 cm of reverse-phase C18 material (YMC*Gel ODS-A, YMC, Allentown, PA). Peptides were gradient-eluted (Solvent A = 0.1 M acetic acid, Solvent

B = 0.1 M acetic acid in 70% v/v acetonitrile, flow rate 200 nl/min) into an LTQ-Orbitrap-Velos or an LTQ-Orbitrap-XL mass spectrometer (Thermo Fisher Scientific) acquiring data-dependent CID fragmentation spectra. In vitro exchange samples were loaded onto an Easy-Spray column (ES800, Thermo Fisher Scientific) and gradient-eluted (Solvent A = 0.1% v/v formic acid in water, Solvent B = 0.1% v/v formic acid in acetonitrile, flow rate 300 nl/min) into an Q Exactive Plus mass spectrometer (Thermo Fisher Scientific) acquiring data-dependent HCD fragmentation spectra. In SILAC experiments comparing inactivated ORF2p catalytic mutants to WT (based on pLD401 [WT], pLD567 [EN⁻], and pLD624 [RT⁻]) peptides were extracted from the gel in two 1 hr incubations with 1.7% v/v formic acid, 67% v/v acetonitrile at room temperature with agitation. Digests were partially evaporated by vacuum centrifugation to remove acetonitrile, and the aqueous component was desalted on Stage Tips. Peptides were loaded onto an Easy-Spray column (ES800, Thermo Fisher Scientific) and gradient-eluted (Solvent A = 0.1% v/v formic acid in water, Solvent B = 0.1% v/v formic acid in acetonitrile, flow rate 300 nl/min) into an Orbitrap Fusion Tribrid mass spectrometer (Thermo Fisher Scientific) acquiring data-dependent fragmentation spectra (either CID spectra alone, or CID and HCD spectra).

## Mass spectrometry data analysis

Raw files were submitted to MaxQuant (*Cox and Mann, 2008*) version 1.5.2.8 for protein identification and isotopic ratio calculation. Searches were performed against human protein sequences (UP000005640, April 2016), custom L1 ORF1p and ORF2p protein sequences, common exogenous contaminants, and a decoy database of reversed protein sequences. Search parameters included fixed modification: carbamidomethyl (C); variable modification: Arg10, Lys8, methionine oxidation; razor and unique peptides used for protein quantitation; requantify: enabled. Contaminants, low-scoring proteins and proteins with one razor+unique peptides were filtered out from the MaxQuant output file 'proteingroups.txt'. The list of contaminants was uploaded from the MaxQuant web-site (http://www.coxdocs.org/; 'contaminants'). Additionally, proteins with the 'POTENTIAL CONTAMINANT' column value '+' were filtered out. Proteins with at least two razor+unique peptides were retained for the analysis. H/(H + L) and L/(H + L) values were derived from unnormalized 'ratio H/L' values and were used for plotting label-swapped RNase-sensitivity and split-tandem data. Unnormalized 'ratio H/L' values were used to calculate H/(H + L) in ORF2p catalytic mutant comparisons and in vitro exchange experiments. These values have been referred to as 'affinities' within the Supplementary Materials. Normalization and clustering procedures applied to data presented in the figures (*Supplementary file 1*) are detailed below and also in Appendix 1. Raw and processed data are available via ProteomeXchange with identifier PXD008542.

To plot RNase-sensitivity affinity capture results (*Figure 1C*), these data were normalized such that proteins that did not change upon treatment with RNases are centered at the origin. The mean value and standard deviation were calculated using the distribution of distances from the origin. The distance threshold for p-value=0.001 was calculated using the R programming language. A circle with radius equal to the threshold was plotted. Points with distances higher than the threshold were marked as black. To plot split-tandem affinity capture results (*Figure 1F*), these data were normalized such that the ORF1p affinity was set to one and the distribution median was maintained. Probabilities associated with selected clusters were calculated based on the frequency distributions of 2- and 3-node clusters present in the data. To plot EN⁻ and RT⁻ mutant affinity capture results (*Figure 4C*), the matrix of detected proteins for each experiment (EN⁻ and RT⁻) was filtered to retain only proteins detected in at least two replicate experiments. The difference between the affinity value of ORF2p and 0.5 value was calculated for each experiment. The affinities of each protein were shifted by the calculated difference. To determine the statistical significance of differentially co-captured proteins between EN⁻ or RT⁻ and WT, respectively, we used a 1-sample t-test and applied Benjamini-Hochberg p-value correction. To determine the statistical significance of differentially co-captured proteins between EN⁻ and RT⁻ we used an unpaired t-test and applied Benjamini-Hochberg p-value correction. To plot in vitro dynamics (*Figure 5B,C*), only proteins which were identified at all time points were used. The cosine similarity method was used to calculate distances between proteins, and hierarchical clustering was used to visualize these distances. To integrate and plot the combined data (*Figure 6*), we calculated Euclidean and cosine distances for each I-DIRT-significant protein pair present in each experiment. Euclidean distances were rescaled to the range (0, 0.9). Proteins not detected in any common experiments were assigned a Euclidian distance of 1 after

rescaling. The total distance between protein pairs was calculated as d = log((rescaled Euclidean distance) * (cosine distance)). This distance was rescaled to the range (0, 1). Hierarchical clustering was used to visualize the calculated distances.

## Gene ontology (GO) analysis

Genes corresponding to the proteins previously reported as significant by I-DIRT (*Taylor et al., 2013*) were tested for statistical overrepresentation using the default settings provided by http://www.panthnerdb.org (*Mi et al., 2017, 2013*), searches were conducted using GO complete molecular function, biological process, and cellular compartment: all results are compiled in *Supplementary file 6*.

## RNA sequencing sample preparation and data acquisition

RNA fractions were obtained from fractions of L1 macromolecules isolated from pLD401 expressing cells by split-tandem affinity capture (*Figure 1D*) and from pLD567 and pLD624 expressing cells by affinity capture (*Figure 4*). The fractions were produced as described above, except few adjustments to favor RNA extraction. Identical stock solutions were used for making buffers but were diluted to working concentration with nuclease-free water (Thermo Fisher Scientific #4387936) and supplemented with RNasin (Promega, Cat.# N2511) – 1:250 during sample extraction and 3xFLAG peptide elution, and 1:1000 during affinity media washing. 600 mg of cell powder was used per preparation, extracted as 3 × 200 mg and pooled after centrifugal clarification, producing ~3 ml of extract. The pooled extracts were combined with magnetic affinity medium from 30 µl of slurry. 75 µl of 1 mg/mL 3xFLAG peptide was used for elution. ½ of the sample was saved for RNA extraction (input) and the other ½ was carried forward to split-tandem IP, using 15 µl α-ORF1 affinity medium slurry. RNAs were extracted from input, α-ORF1 supernatant fractions, as well as directly from the α-ORF1 affinity medium (elution) with 500 µl of TRIzol (Thermo Fisher Scientific #15596026), following the manufacturer's instructions. Aqueous TRIzol extracts were re-extracted in an equal volume of chloroform, and the aqueous phase was again removed; 1 µl (~15 ug) of GlycoBlue (Thermo Fisher Scientific #AM9516) and 2 ul of RNasin were added to this and mixed before combining with 250 µl of isopropanol and incubating for 10' on ice to precipitate RNA. Alcohol precipitates were centrifuged at 20 k RCF for 30' @ 4°C and the pellets were washed twice with 500 µl of cold 70% ethanol, then air dried for 5' at RT and re-solubilized in 100 µl of nuclease-free water. Extracted RNAs in water were then further purified and concentrated using a Qiagen RNeasy MinElute Cleanup Kit (#74204) following the manufacturer's instructions, and eluted in 14 µl of nuclease-free water. 5 µl of purified RNA was used directly in RNA fragmentation. Libraries were prepared with unique barcodes and were pooled at equimolar ratios. The pool was denatured and sequenced on Illumina NextSeq 500 sequencer using high output V2 reagents and NextSeq Control Software v1.4 to generate 75 bp single reads, following manufacturer's protocols (#15048776, Rev.E).

## RNA sequencing data analysis

Human genome hg19 GRCh37.87 (FASTA) and annotation (GTF file) were downloaded from ENSEMBL (ftp://ftp.ensembl.org/pub/grch37/release-90) and reference FASTA and GTF files were created by combining the human genome and *ORFeus*-Hs from pLD401 (*Taylor et al., 2013*); *Supplementary file 7*: ORFeus-Hs_pLD401.gbk). To map sequencing reads onto the reference genome and produce differential gene expression analysis: (1) FASTAQ files were trimmed via trimmomatic (*Bolger et al., 2014*) using the following parameters: -phred33 -threads 8, LEADING:3 TRAILING:3 SLIDINGWINDOW:4:16 MINLEN:25; (2) mapping was performed via STAR (*Dobin et al., 2013*) version 2.5.3a (https://github.com/alexdobin/STAR) using the following parameters: -runThreadN 8, –quantMode GeneCounts, –outSAMtype BAM SortedByCoordinate, –outFilterMatchNmin 30; (3) the results were output to one binary alignment map file for each sample matched to the reference; (4) genes with the coverage of 10 or more reads in at least three experiments were selected; and (5) data was normalized using the 'DESeq2' (*Love et al., 2014*) R package version 1.14.1. Raw and normalized mapped, annotated reads are described in *Supplementary file 4*. FASTAQ files are available through Gene Expression Omnibus at NCBI: GSE108270.

## L1 element amplification protocol (LEAP)

We generated an N-terminally acetylated, C-terminally amidated version of the ORF1p peptide (MENDFDELRE) as a di-peptide composed of repeats of the same sequence linked by a four-unit polyethylene glycol moiety; which was used to elute ORF1p-containing complexes from α-ORF1p medium at a concentration of approximately 2 mM (Appendix 1; *Supplementary file 4*). Peptides were synthesized by standard Fmoc solid-phase synthesis methods (*Kates and Albericio, 2000*); the incorporation of a PEG spacer into the peptide sequence was accomplished using N-Fmoc-amido-(PEG) n-acid building blocks. 400 mg of cryogenically milled L1-expressing cells (pLD401 and pLD561) were subjected to split-tandem affinity capture as described above, but with native elution from α-ORF1p medium and included the addition of RNasin (Promega #N2515) at 1:500 v/v to the extraction buffer; 1x protease inhibitors and 1:200 v/v RNasin were also added to the 3xFLAG peptide and ORF1p-derived di-peptide solutions. For α-FLAG affinity capture, competitive elution was achieved using 60 µl of 1 mg/ml 3xFLAG peptide. Of this, 20 µl were held aside (Input), 40 µl were carried forward to α-ORF1p affinity capture. The ORF1p-depleted fraction was retained (Sup) and the captured material was eluted with 40 µl ORF1p di-peptide (Elu). Half of each fraction (Input, Sup, Elu) was set aside for protein analysis (*Supplementary file 4*) and to the other half, glycerol was added to 25% v/v (using a 50% v/v glycerol solution); the latter were subsequently analyzed for enzymatic activity by LEAP. Raw data resulting from these assays is located in *Supplementary file 4*. For LEAP, 2 µl from each of the above-described fractions were used in a 50 µl reaction, and 1 µl of each LEAP assay was used in SYBR Green qPCR (carried out in triplicate) as previously described (*Taylor et al., 2013*). As controls, (1) an untagged L1RP construct was used in a 'mock purification,' and (2) pLD401-derived 'Input' was heated at 100°C for 5 min and then added to the reaction mix, respectively. Neither produced detectable activity (*Supplementary file 4*). A second LEAP analysis was later carried out on an independently prepared set of fractions, prepared as above, stored frozen −80°C in 25% v/v glycerol.

## ORF protein immunofluorescence analysis in HeLa cells

Tet-on HeLa M2 cells (*Hampf and Gossen, 2007*) (a gift from Gerald Schumann), were transfected and selected with 1 µg/ml puromycin for three days. Puromycin-resistant cells were plated on coverslips pre-coated for 1–2 hr with 10 µg/ml fibronectin in PBS (Life Technologies). 8–16 hr after plating, L1 was induced with 1 µg/ml doxycycline. 24 hr later, cells were fixed in 3% paraformaldehyde for 10 min. Fixative was then quenched using PBS containing 10 mM glycine and 0.2% w/v sodium azide (PBS/gly). The cells were permeabilized for 3 min in 0.5% Triton X-100 and washed twice with PBS/gly. Staining with primary and secondary antibodies was done for 20 min at room temperature by inverting coverslips onto Parafilm containing 45 ml drops of PBS/gly supplemented with 1% BSA, mouse α-FLAG M2 (Sigma, 1:500), rabbit α-ORF1 JH73 (1:4000) (*Taylor et al., 2013*), Alexa Fluor 488 conjugated α-mouse IgG (Life Technologies, 1:1000), and Alexa Fluor 568 conjugated α-rabbit IgG (Life Technologies, 1:1000). DNA was stained prior to imaging with Hoechst 33285 (Life Technologies, 0.1 µg/ml). Epifluorescent images were collected using an Axioscop microscope (Zeiss, Jena, Germany) equipped for epifluorescence using an ORCA-03G CCD camera (Hamamatsu, Japan).

## ORF2p+ nuclei proximity analysis

For each microscope field, nuclei were identified and spatially located using a custom script in ImageJ, consisting of Otsu thresholding and watershed transformation of DAPI signal to segment each of the nuclei. ORF2p positive nuclei were differentiated from ORF2p negative nuclei by using another thresholding script for the ORF2p fluorescence channel and cross-registering the associated nuclei; all ORF2p positive nuclei were then hand-verified and then coordinates were converted into microns. The number of ORF2p+ nuclei per field, x, and a corresponding random distribution of x nuclei was calculated by randomly and repeatedly (n = 1000) selecting x nuclei among all nuclei. The random distribution was used to calculate Bonferroni corrected p-values for the pairwise distances between ORF2p+ nuclei. The distribution of ORF2p+ inter nuclei distances was then compared to the distribution of random inter-nuclei distances using Welch's t-test. The custom scripts used to select nuclei and calculate statistics, extracted data, calculated distances, p-values, and raw images are presented in the supplement (*Supplementary file 5*; *Figure 3—source data 1*).

## Acknowledgements

This work was supported in part by National Institutes of Health (NIH) grants P41GM109824 (to MPR), P41GM103314 (to BTC), P50GM107632 (to JDB), and R01GM126170 (to JL), and by the 5-100 Russian Academic Excellence Program. The mass spectrometric analysis of proteins co-captured with chromatin associated ORF2p (*Supplementary file 3*) was conducted within the NYU School of Medicine Proteomics Resource Lab, which is partially supported by the Laura and Isaac Perlmutter Cancer Center Support Grant, NIH P30CA16087, and NIH 1S10OD010582. RNA sequence library preparation and next-generation sequencing was carried out by The Rockefeller University Genomics Resource Center. Peptide synthesis was performed by Henry Zebroski at The Rockefeller University Proteomics Resource Center. We thank Carolyn Machamer for advice and resources supporting fluorescence microscopy, and Lixin Dai for assistance with LEAP assays. This paper is subject to the NIH Public Access Policy. The authors declare no conflicts of interest.

## Additional information

### Funding

| Funder | Grant reference number | Author |
| --- | --- | --- |
| National Institutes of Health | P41GM103314 | Brian T Chait |
| National Institutes of Health | P41GM109824 | Michael P Rout |
| National Institutes of Health | P50GM107632 | Jef D Boeke |
| National Institutes of Health | R01GM126170 | John LaCava |

The funders had no role in study design, data collection and interpretation, or the decision to submit the work for publication.

### Author contributions

Martin S Taylor, Conceptualization, Investigation, Visualization, Methodology, Writing—original draft, Writing—review and editing; Ilya Altukhov, Data curation, Software, Formal analysis, Visualization, Methodology; Kelly R Molloy, Data curation, Investigation, Methodology; Paolo Mita, Investigation, Writing—review and editing; Hua Jiang, Emily M Adney, Aleksandra Wudzinska, Investigation; Sana Badri, Software, Formal analysis; Dmitry Ischenko, Formal analysis; George Eng, Software; Kathleen H Burns, Michael P Rout, Resources, Funding acquisition, Writing—review and editing; David Fenyö, Resources, Formal analysis; Brian T Chait, Resources, Funding acquisition; Dmitry Alexeev, Resources, Data curation, Formal analysis, Supervision, Funding acquisition; Jef D Boeke, Conceptualization, Resources, Funding acquisition, Writing—review and editing; John LaCava, Conceptualization, Data curation, Supervision, Investigation, Visualization, Methodology, Writing—original draft, Project administration, Funding Acquisition, Writing—review and editing

### Author ORCIDs

Martin S Taylor http://orcid.org/0000-0001-5824-142X
Ilya Altukhov https://orcid.org/0000-0001-9821-1890
Paolo Mita http://orcid.org/0000-0002-2093-4906
Kathleen H Burns https://orcid.org/0000-0003-1620-3761
David Fenyö http://orcid.org/0000-0001-5049-3825
Dmitry Alexeev http://orcid.org/0000-0003-0783-1176
Jef D Boeke http://orcid.org/0000-0001-5322-4946
John LaCava http://orcid.org/0000-0002-6307-7713

### Decision letter and Author response

Decision letter https://doi.org/10.7554/eLife.30094.036
Author response https://doi.org/10.7554/eLife.30094.037

# Additional files

## Supplementary files

• Supplementary file 1. Supporting and supplemental data for the figures and experiments: RNase – *Figure 1C*; Tandem – *Figure 1F*; Mutants – *Figure 4*; Exchange – *Figure 5*; Exchange distance matrix – *Figure 5C*; Integration – *Figure 6*; Integration distance matrix – *Figure 6B*; Raw – unnormalized values extracted from the MaxQuant proteinGroups.txt file for the experiments presented in this study.

DOI: https://doi.org/10.7554/eLife.30094.013

• Supplementary file 2. Supporting and supplemental data for overexpression and siRNA knockdown experiments.

DOI: https://doi.org/10.7554/eLife.30094.014

• Supplementary file 3. Supporting and supplemental data for affinity capture of ORF2p-3xFLAG from fractionated chromatin.

DOI: https://doi.org/10.7554/eLife.30094.015

• Supplementary file 4. Supporting and supplemental data for figures and experiments: RNA sequencing and LEAP assays – *Figure 2*.

DOI: https://doi.org/10.7554/eLife.30094.016

• Supplementary file 5. Supporting and supplemental data for Nuclear Location and ORF2p status: *Figure 3*.

DOI: https://doi.org/10.7554/eLife.30094.017

• Supplementary file 6. Supporting and supplemental data for GO analysis.

DOI: https://doi.org/10.7554/eLife.30094.018

• Supplementary file 7. ORFeus-Hs sequence from pLD401 included in our reference FASTA file: *Figure 2* and *Supplementary file 4*.

DOI: https://doi.org/10.7554/eLife.30094.019

• Transparent reporting form

DOI: https://doi.org/10.7554/eLife.30094.020

## Major datasets

The following datasets were generated:

| Author(s) | Year | Dataset title | Dataset URL | Database, license, and accessibility information |
|---|---|---|---|---|
| John LaCava | 2017 | Dissection of affinity captured LINE-1 macromolecular complexes | http://proteomecentral.proteomexchange.org/cgi/GetDataset?ID=PXD008542 | Publicly available at ProteomeXchange (accession no: PXD008542) |
| John LaCava | 2017 | RNAs associated with affinity captured LINE-1 ribonucleoproteins | https://www.ncbi.nlm.nih.gov/geo/query/acc.cgi?acc=GSE108270 | Publicly available at the NCBI Gene Expression Omnibus (accession no: GSE108270) |

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

# Appendix 1

DOI: https://doi.org/10.7554/eLife.30094.021

All normalized affinity values, derived from H/(H + L) and L/(H + L) isotopic ratios, can be found in *Supplementary file 1* on the appropriate sheet; pre-normalization values are located on the sheets named 'Integration' and 'Raw.'

## Modified SILAC strategy

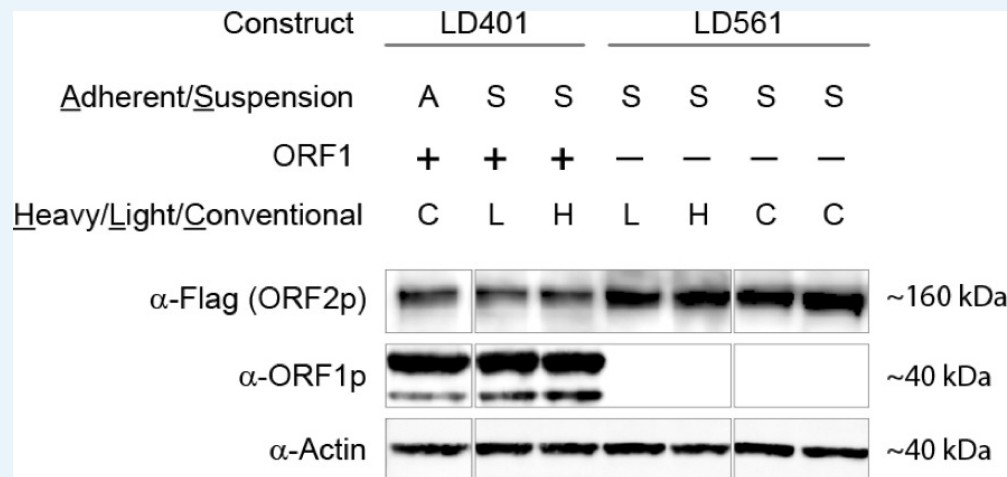

**Appendix 1—figure 1.** SILAC suspension expression of L1 constructs: western blotting.

DOI: https://doi.org/10.7554/eLife.30094.022

### SILAC suspension expression of L1 constructs

Western blotting of cells grown in adherent culture with puromycin selection (A) or suspension culture with transient transfection (S). Cells were grown in heavy isotope-supplemented media ($^{13}C$ $^{15}N$ lysine and arginine) (H), light isotope-supplemented media (L) or conventional commercial media (C) supplemented with tetracycline-free serum and L-glutamine. Note that serum used for heavy and light growth is dialyzed to remove amino acids. Construct LD401: synthetic ORFeus-HS, full L1 coding sequence (both ORFs and 3'UTR) with ORF2-3xFlag. Construct LD561: identical except for the absence of ORF1.

## RNase sensitivity affinity capture

### Data normalization

The RNase sensitivity data were rescaled and normalized such that proteins that did not change upon treatment with RNases were centered at the origin and those that were completely sensitive would give a value of 1.0. In a perfect experiment, unchanging proteins would yield a ratio of 0.5 when comparing the fraction of each protein present in the BSA-treated sample to the sum of both the BSA- and RNase-treated samples; i.e. 1 / (1 + 1). However, our data show some variability (below, left and also *Supplementary file 1*), with one replicate centering on ~0.4 (red) and another ~0.6 (blue). Therefore, we normalized the data such that the peaks at ~0.4 and~0.6 were both re-centered at 0.5. From this set, 0.5 was subtracted from the data (centering insensitive proteins at the origin, and completely sensitive proteins at 0.5), followed by multiplication by two to expand the data to cover the range from 0 (insensitive) to 1 (completely sensitive); depicted below, right. These latter two

transformations are encompassed by the functions: g(x)=x + b [where b = −0.5] and f(g(x))=a (x + b) [where a = 2].

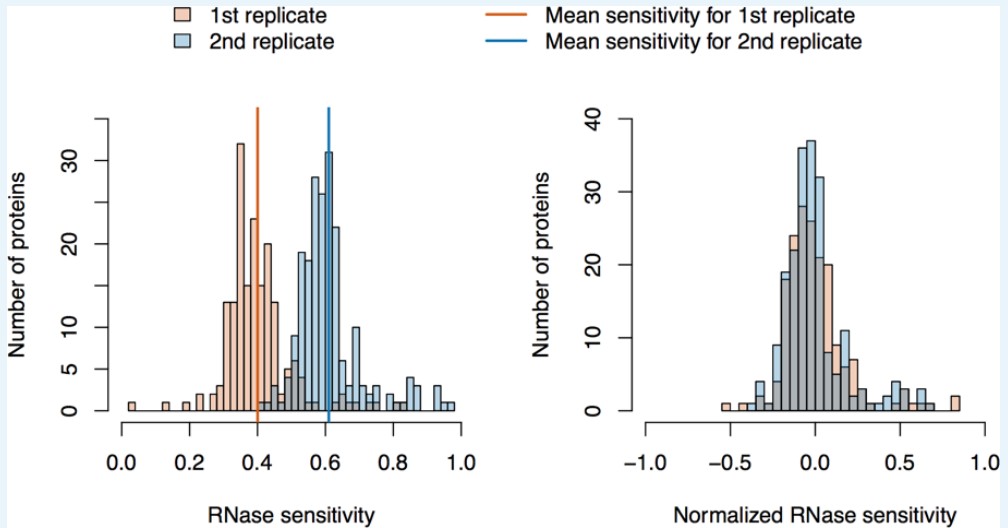

**Appendix 1—figure 2.** RNase sensitivity affinity capture: data normalization.
DOI: https://doi.org/10.7554/eLife.30094.023

## RNase normality test

The distances from the (0,0) point to protein coordinates were calculated. Proteins with distance less than two median distances were selected. The Shapiro-Wilk normality test (the null-hypothesis of this test is that the population is normally distributed) was applied for the distances (p-value=0.29). The distribution of the distances was plotted as a histogram displaying the frequency (y-axis) versus RNase sensitivity (x-axis) of a simulation of normally distributed data (shown in black) and the actual data (*Supplementary file 1*) shown in blue. A Q-Q plot was also drawn.

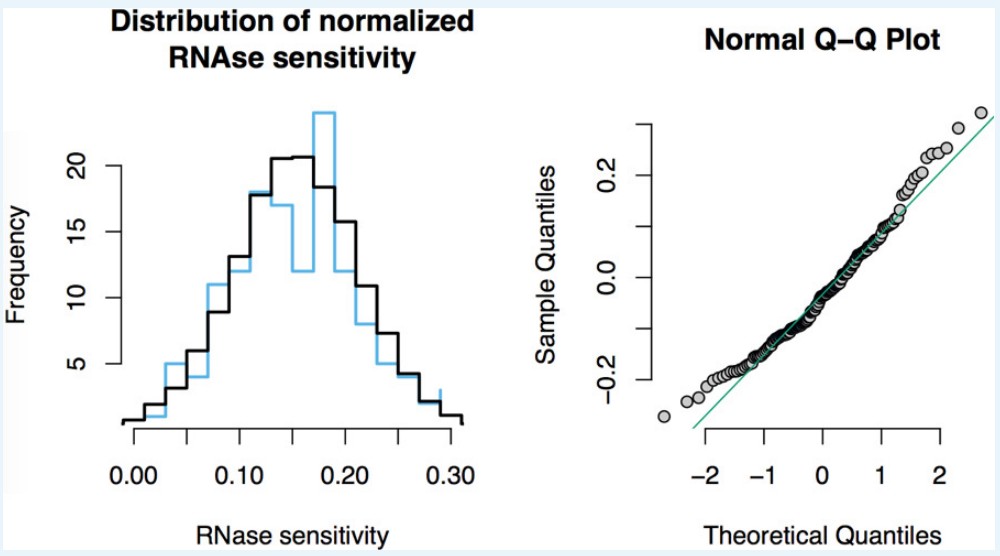

**Appendix 1—figure 3.** RNase sensitivity affinity capture: normality test.
DOI: https://doi.org/10.7554/eLife.30094.024

## Split-tandem affinity capture

### Data normalization

The data were treated as follows: a and b coefficients were calculated as solutions of *equation 1*; the normalized values were calculated using the *equation 2*.

$$\begin{pmatrix} a \\ b \end{pmatrix} \begin{pmatrix} \text{median} & 1 \\ \text{ORF1} & 1 \end{pmatrix} = \begin{pmatrix} \text{median} \\ 1 \end{pmatrix} \tag{1}$$

$$x_{\text{normalized}} = (a^* x_{\text{initial}}) + b \tag{2}$$

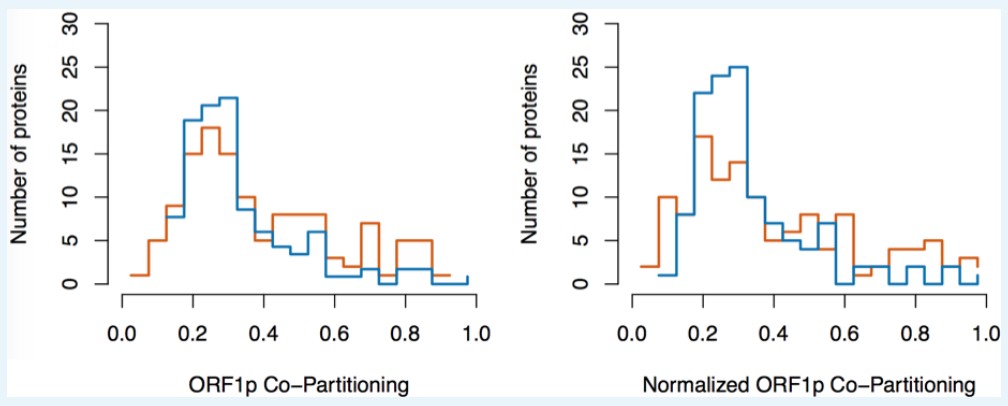

**Appendix 1—figure 4.** Split-tandem affinity capture: data normalization.
DOI: https://doi.org/10.7554/eLife.30094.025

## Calculate the distances between node pairs

Distance between two points A and B with coordinates $(A_x, A_y)$ and $(B_x, B_y)$ was calculated as:

$$\sqrt{\left(A_x - B_x\right)^2 + \left(A_y - B_y\right)^2}$$

For each three points, the mean paired distance was calculated. The distributions of mean values are presented in the histograms below.

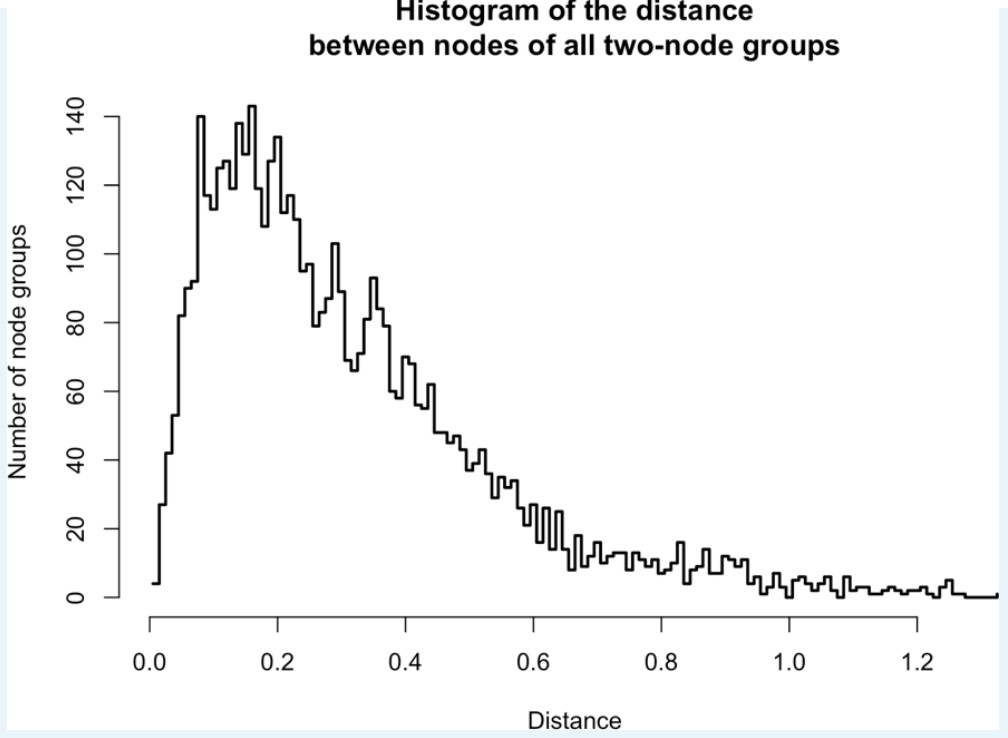

**Appendix 1—figure 5.** Distances between two-node groups.
DOI: https://doi.org/10.7554/eLife.30094.026

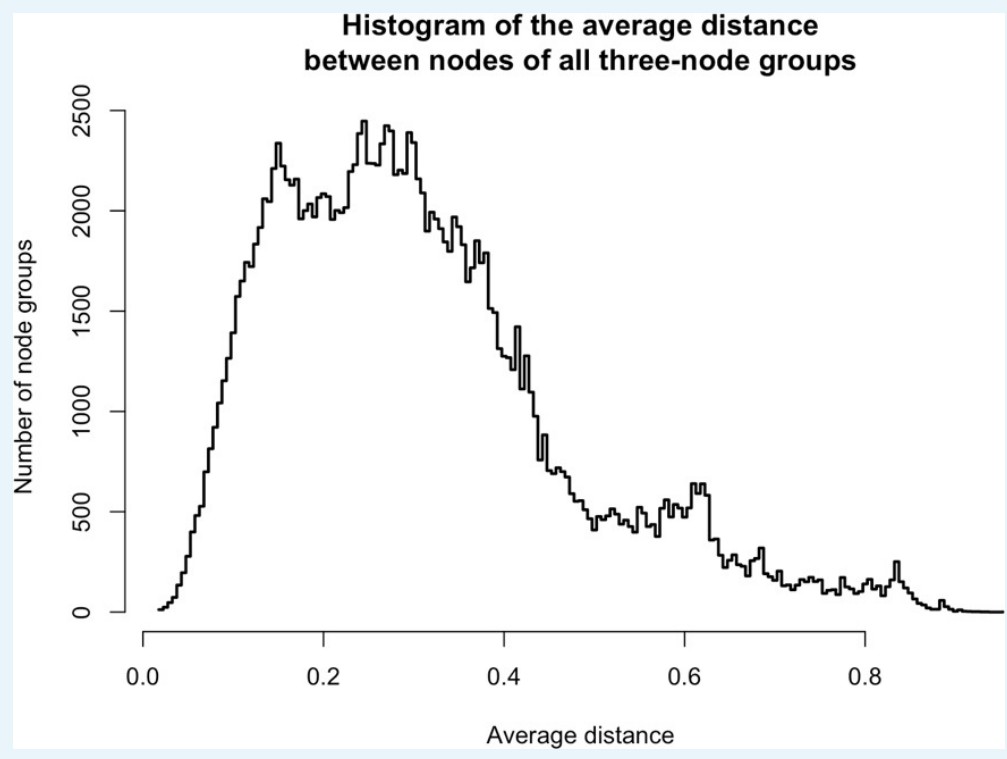

**Appendix 1—figure 6.** Distances between three-node groups.
DOI: https://doi.org/10.7554/eLife.30094.027

## Associated likelihoods of selected clusters

Here, likelihood is defined as the frequency with which the same mean distance or less is observed within the distribution of clusters with the same number of nodes (above).

PURA/PURB/PCNA: Likelihood = $3.2 \times 10^{-7}$
PABPC1/PABPC4: Likelihood = 0.0008388427
HSPA8/HSPA1A: Likelihood = 0.0001991309
NAP1L1/IPO7: Likelihood = 0.0075885198

## Efficacy elution from α-ORF1 4H1 affinity medium using ORF1p peptides

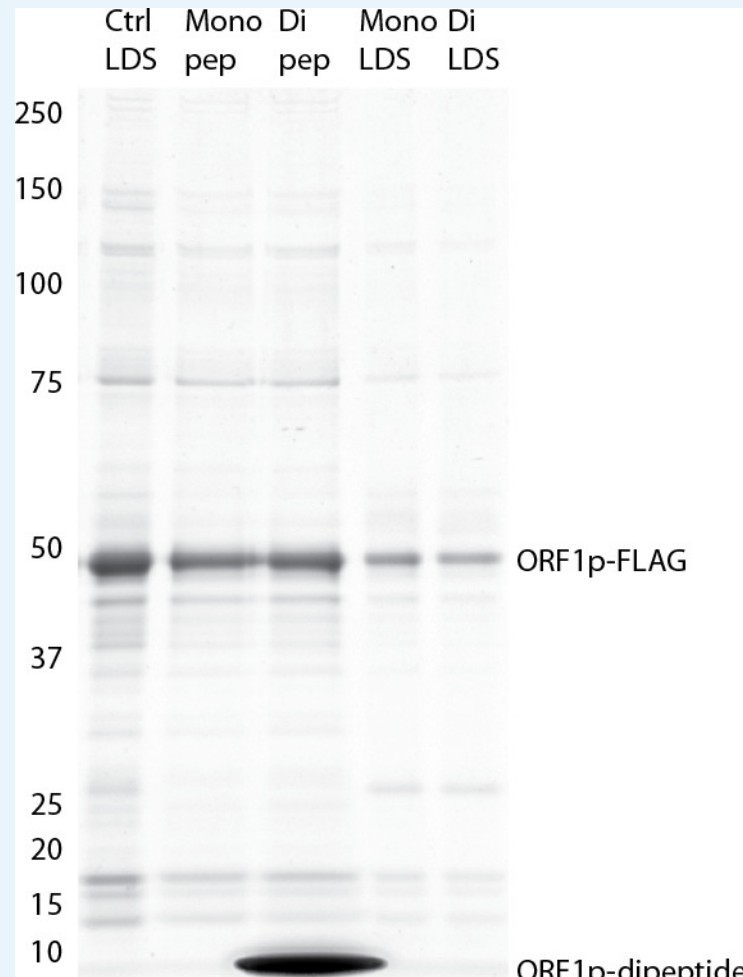

**Appendix 1—figure 7.** Efficacy of elution using ORF1p peptides: Coomassie blue stained gel .
DOI: https://doi.org/10.7554/eLife.30094.028

ORF1p-FLAG was purified from 25 mg of cryo-milled HEK-293T$_{LD}$ expressing pLD288 using α-ORF1 affinity medium, essentially as previously described (*Taylor et al., 2013*), and then eluted either eluted directly with 15 μl of 1x LDS, 70°C for 5 min (Ctrl LDS), with 2 mM monomeric ORF1 peptide (Mono pep), or 2 mM dimeric ORF1 peptide (Di pep) (in both cases for 15 min at room temperature). After elution with peptide, the affinity medium was further eluted with 1x LDS at 70°C for 5 min (Mono and Di LDS, respectively).

## Retrotransposition mutants affinity capture

The distributions of normalized affinities for the two sets of experiments are shown below.

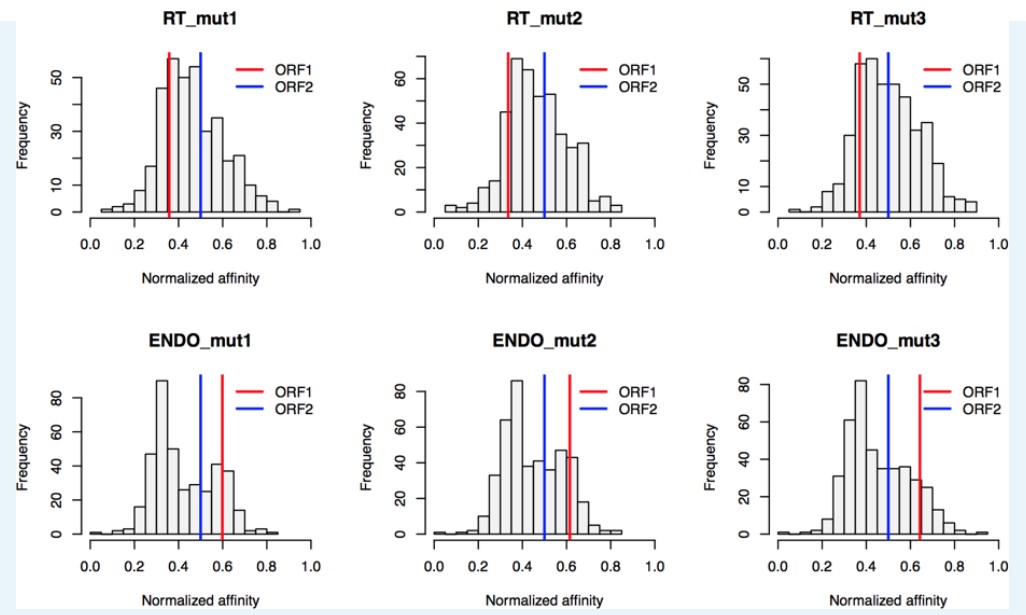

**Appendix 1—figure 8.** Retrotransposition mutants affinity capture: distributions of normalized affinities.

DOI: https://doi.org/10.7554/eLife.30094.029

## Protein in vitro exchange

The distributions of H/(H + L) values present at each time point are shown.

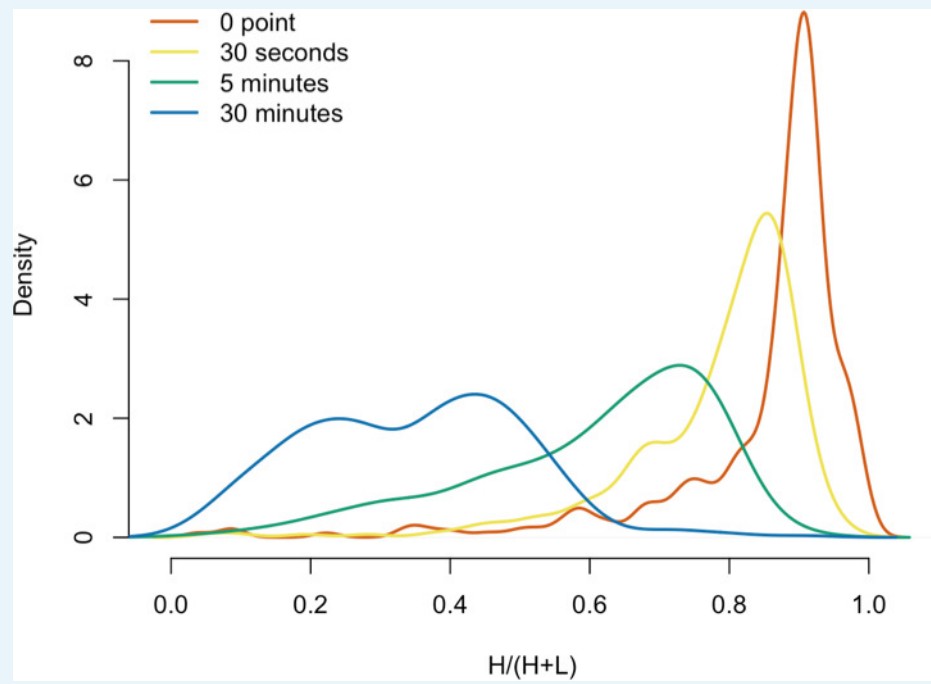

**Appendix 1—figure 9.** Protein in vitro exchange: the distributions of H/(H+L) values.

DOI: https://doi.org/10.7554/eLife.30094.030

# Affinity capture of ORF2p-3xFLAG L1 from fractionated chromatin and MS analyses

## Cell culture

Briefly, suspension grown HEK-293TLD cells were seeded at $1 \times 10^6$ cells/ml in 100 ml of medium and transfected with pLD401 (ORF2p-3xFLAG L1 construct) or pLD259 (untagged L1 control construct) plasmid DNA. The transfection mixture consisted of Hybridoma serum free media (1/20 of final volume), PEI (3 µg/ml final volume) and plasmid DNA (1 µg/ml final volume). The mixture was incubated for 15 min at room temperature before adding to cell suspension. 24 hr post transfection, cells were split 1:3 into 1 µg/ml puromycin media. Expression was induced 48 hr post transfection by the addition of doxycycline (1 µg/ml) and maintained for 48 hr before collection for chromatin fractionation. A total of 900 ml final cell suspension per construct ($\approx 3 \times 10^6$ cells/ml) were prepared as follows.

## Chromatin fractionation

Cell suspensions were centrifuged at 200 RCF for 10 min and washed with 20 ml PBS. Cell pellets were resuspended in 5 ml of Buffer A (100 mM HEPES, 1.5 mM MgCl2, 0.34 M sucrose, 10% (v/v) glycerol; with 1 mM DTT and protease inhibitors freshly added). Triton X-100 was added to 0.1% (v/v) final concentration and cells were allowed to swell on ice for 10 min. Nuclei were pelleted for 5 min at 1300 RCF, 4°C and the supernatant (cytoplasmic fraction) was discarded. Nuclei were resuspended in 2.5 ml Buffer B (3 mM EDTA, 0.2 mM EGTA; with 1 mM DTT and protease inhibitors freshly added) and incubated on ice for 30 min before centrifuging at 1700 RCF for 5 min. The soluble nuclear fraction was discarded and the insoluble material was washed twice with Buffer B. The remaining chromatin fraction was resuspended in 5 ml MNase buffer (a Tris buffered 10 mM KCl, 1 mM CaCl2 solution) supplemented with 5 U/ml micrococcal nuclease and incubated at 37°C for 5 min with agitation. The reaction was quenched by adding EGTA to 1 mM final concentration and incubating for 2 min. The solution was centrifuged for 5 min at max speed and supernatant (chromatin fraction) transferred to a fresh tube.

## Immunoprecipitation

The chromatin fractions were normalized by Bradford Assay and equal amounts of proteins were used for the IP. The chromatin fractions were diluted in concentrated buffer to a final concentration of 500 mM NaCl, 20 mM HEPES, pH 7.4, and 1% (v/v) Triton X-100 (same formula used as washing buffer, below). 50 µl of magnetic beads (Life Technologies 14311D) conjugated to FLAG-M2 antibody (Sigma F1804) were added to the fractions incubated for 1 hr at 4°C under end-over-end rotation. The affinity media were washed 10 times with washing and twice with 500 mM NaCl, 20 mM HEPES, pH 7.4, and 0.1% (v/v) Triton X-100. Proteins were eluted for 30 min at room temperature under continuous shaking in 50 µl of 1 mg/ml 3xFLAG peptide (Sigma F4799) diluted in washing buffer with 0.1% Triton X-100. The eluates were collected and combined with NuPAGE 4x LDS Sample Buffer (Novex) to a final concentration of 1x.

## Preparation for mass spectrometry

The samples were reduced with 2 µl of 0.2M dithiothreitol (Sigma) for one hour at 57°C at pH 8.0. Next the samples were alkylated with 2 µl of 0.5M iodoacetamide (Sigma) for 45 min at room temperature in the dark. The samples were loaded on a NuPAGE 4–12% Bis-Tris Gel 1.0 mm (Life Technologies) and run for 6 min at 200V. The gel was stained with GelCode Blue Stain Reagent (Thermo). The gel plugs were excised and destained for 15 min in a 1:1 (v/v) solution of methanol and 100 mM ammonium bicarbonate. The buffer was exchanged and the samples were destained for another 15 min. This was repeated for another three cycles. The gel plugs were dehydrated by washing with acetonitrile, and then further dried by placing in a

SpeedVac for 20 min. The gel plugs were treated with 250 ng of sequencing grade modified trypsin (Promega) by adding directly on top of the dried gel plugs, and then enough 100 mM ammonium bicarbonate was added in order to cover the gel pieces. The gel plugs were allowed to shake at room temperature and digestion proceeded overnight. The digestion was halted by adding a slurry of R2 50 µm Poros beads (Applied Biosystems) in 5% formic acid and 0.2% trifluoroacetic acid (TFA) to each sample at a volume equal to that of the ammonium bicarbonate added for digestion. The samples were allowed to shake at 4$^0$C for three hours. The beads were loaded onto C18 ziptips (Millipore), equilibrated with 0.1% TFA, using a microcentrifuge for 30 s at 6,000 rpm. The beads were washed with 0.5% acetic acid. Peptides were eluted with 40% acetonitrile in 0.5% acetic acid followed by 80% acetonitrile in 0.5% acetic acid. The organic solvent was removed using a SpeedVac concentrator and the sample reconstituted in 0.5% acetic acid.

## Mass spectrometry analysis – Thermo Orbitrap Elite instrument

An aliquot of each sample was loaded onto an Acclaim PepMap100 C18 75 µm x 15 cm column with 3 µm bead size, coupled to an EASY-Spray 75 µm x 50 cm PepMap C18 analytical HPLC column with a 2 µm bead size, using the auto sampler of an EASY-nLC 1000 HPLC (ThermoFisher) and solvent A (2% acetonitrile, 0.5% acetic acid). The peptides were eluted into a ThermoFisher Scientific Orbitrap Elite Hybrid Ion Trap Mass Spectrometer increasing from 2% to 30% solvent B (90% acetonitrile, 0.5% acetic acid) over 60 min, followed by an increase from 30% to 40% solvent B over 30 min. Solvent B was then put to 100% and held at 100% for 20 min. High resolution full MS spectra were obtained with a resolution of 60,000 at 400 m/z, an AGC target of 1e6, with a maximum ion time of 200 ms, and a scan range from 300 to 1500 m/z. Following each full MS scan, fifteen data-dependent MS/MS spectra were acquired. The MS/MS spectra were collected in the ion trap, with an AGC target of 1e4, maximum ion time of 150 ms, one microscan, 2 m/z isolation window, fixed first mass of 150 m/z, and Normalized Collision Energy (NCE) of 35.

## Mass spectrometry analysis – Thermo Fusion instrument

An aliquot of each sample was loaded onto an Acclaim PepMap100 C18 75 µm x 15 cm column with 3 µm bead size, coupled to an EASY-Spray 75 µm x 50 cm PepMap C18 analytical HPLC column with a 2 µm bead size, using the auto sampler of an EASY-nLC 1000 HPLC (ThermoFisher) and solvent A (2% acetonitrile, 0.5% acetic acid). The peptides were eluted into a ThermoFisher Scientific Orbitrap Fusion Mass Spectrometer increasing from 2% to 30% solvent B (90% acetonitrile, 0.5% acetic acid) over 60 min, followed by an increase from 30% to 40% solvent B over 30 min. Solvent B was then put to 100% and held at 100% for 20 min. High resolution full MS spectra were obtained with a resolution of 120,000, an AGC target of 400,000, with a maximum ion time of 50 ms, and a scan range from 400 to 1500 m/z. The MS/MS spectra were collected in the ion trap, with an AGC target of 100, maximum ion time of 250 ms, one microscan, 2 m/z isolation window, fixed first mass of 150 m/z, and Normalized Collision Energy (NCE) of 27.

## Data processing

All acquired MS2 spectra were searched against a UniProt human database using Sequest within Proteome Discoverer (ThermoScientific). The search parameters were as follows: precursor mass tolerance ±10 ppm, fragment mass tolerance ±0.4 Da, digestion parameters allowing trypsin two missed cleavages, fixed modification of carbamidomethyl on cysteine, variable modification of oxidation on methionine, and variable modification of deamidation on glutamine and asparagine. The results were filtered to only include proteins identified by at least two peptides.

