## [Decision Letter]

Thank you for submitting your article "Dissection of purified LINE-1 reveals distinct nuclear and cytoplasmic intermediates" for consideration by *eLife*. Your article has been reviewed by three peer reviewers, and the evaluation has been overseen by a Reviewing Editor and James Manley as the Senior Editor. The following individuals involved in review of your submission have agreed to reveal their identity: Geoffrey Faulkner (Reviewer #2); Victoria Belancio (Reviewer #3).

The reviewers have discussed the reviews with one another and the Reviewing Editor has drafted this decision to help you prepare a revised submission.

Summary:

This paper reports the protein content of LINE-1 complexes in the cytoplasm and nucleus of cells undergoing retrotransposition events, and uncovers an array of distinctive complexes. The reviewers uniformly found the story to be of much interest and to constitute a significant advance beyond published work. They are enthusiastic.

We include all their reviews here for consideration. There were raised many shared issues of what were judged to be overinterpretations or overstatements of the data. I think most of the comments can be addressed by rewriting.

A common thread is the point that we cannot know whether the complexes are on a direct pathway to retrotransposition, or whether they are even on a pathway at all. Some disclaimers here are needed.

Essential revisions:

*Reviewer 1:*

The point about Orf2 traversing the RNA in the cytoplasm, given what we think we know about the process of RT here, seems cogent. Here it is probably correct that the data do not rule out the presence of Orf1 in the nucleus at the time of RT. This needs to be addressed.

*Reviewer 2:*

There are real issues about the cell localization data. We should not assume that the localization of cellular proteins is according to the dogma in non-transposing cells, outside of the Orf complexes. There are real issues about IF quantification that need to be addressed.

*Reviewer 3:*

Almost all the comments here are valid and readily addressed. Again, the issue of Orf1 in the nucleus is raised as a red flag that needs to be addressed, likely by softening of claims.

We look forward to receiving a revised version with a point-by-point response to the reviews.

*Reviewer #1:*

This study provides a significant advance in our understanding of biochemical interactions of LINE-1 encoded proteins, with solid evidence for multiple, varied interactions with distinct host cell components in both the nucleus and the cytoplasm. The authors employ sophisticated, quantitative methods to interrogate protein interactions of (primarily) the LINE-1 ORF2 protein, thereby gaining new insight into the complexity of the host-parasite relationship and process of LINE-1 retrotransposition. It represents a significant step forward for the field, building on results of Taylor published in 2013 in Cell. There are however some issues with the wording of some conclusions overstepping the limits of what is documented by the data presented, as specifically noted below. With wording changes to more accurately describe/interpret the data the most serious limitation of this work (the lack of a co-analysis of the L1 nucleic acid intermediates), could be considered minor.

In Abstract "our findings support the presence of multiple L1-derived retrotransposition intermediates in vivo"; oversteps the actual data because there is no direct evidence to address which of these various complexes are bona fide retrotransposition intermediates vs which reflect the various ways the cell interferes with/prevents retrotransposition.

In numerous instances the term "purified" is used when the data support "enrichment" but not purification. This leads to confusion when the "purified" complex turns out to be, as is the point of the work, multiple distinct complexes.

The conclusion "[…]ORF1p probably does not interact with ORF2p in the nucleus", is not supported by the data, which simply show that there are RNA- and ORF1p-independent interactions between ORF2p and non-L1 cellular proteins in the nucleus. Agreed that the data show a persistent complex of ORF2p that lacks ORF1p, but that is not the same as the above. There is likewise no data to support "ORF1p may be displaced from the L1 RNA during the nuclear portion of the lifecycle" – maybe there is also no remaining L1 RNA by the time the ORF2 complexes lacking ORF1p are formed. The authors should provide supporting data to support this claim or remove it.

Subsection “4.4. The effects of retrotransposition-blocking point mutations on the interactomes of purified L1 RNPs”. TPRT is target primed reverse transcription, not "template".

Subsection “4.4. The effects of retrotransposition-blocking point mutations on the interactomes of purified L1 RNPs”. What evidence suggests that ORF2 traverses L1 RNA in cytoplasm? Given that this RT gets its primer from the target DNA this comment seems unwarranted. This unnecessary speculation is not appropriate for the Results section. In fact the effects of the RT and EN mutations are actually so subtle (~2x) compared to the robust effects they have on retrotransposition (>>100X) it may speak more to the fact that neither of these most abundant L1-containing protein complexes (both nuclear and cytoplasmic) are on the productive retrotransposition pathway.

Figure 6 model – there is no evidence linking the L1 RNA to this nuclear complex with ORF2p and without ORF1p. Given that ORF1p is required to get ORF2p into the nucleus, that some ORF1p does get to the nucleus and in a form that is somehow unique (based on different reactivity of the various antibodies), and that 25% of the ORF2p follows ORF1p in the sequential enrichments, an important role for ORF1p in the nucleus seems highly likely. It would be reasonable from the data presented to consider at least two nuclear forms with ORF2p containing complexes, one with the L1 RNA and ORF1p and one without these other L1 components but with the other nuclear components.

*Reviewer #2:*

Molloy et al. perform affinity purifications followed by quantitative mass spectrometry to identify proteins that interact with the L1-encoded proteins and mRNA in ribonucleoprotein particles (RNPs). Their results suggest the existence of multiple different L1 RNPs that contain the L1-encoded proteins ORF1p and ORF2p and various cellular factors at distinct stoichiometries and in discrete subcellular locations. The major advance of this manuscript over the work of Taylor et al. (Cell, 2013) is the resolution of the subcellular localisation of the different RNP intermediates. These data are really interesting and should be the starting point for a range of follow-up investigations. However, the strength with which some of the conclusions are presented is not sufficiently justified by the available data and this should be addressed prior to publication.

1) Most of the conclusions about the subcellular localisation of ORF1p/ORF2p versus ORF2p-only RNP complexes are drawn from the known subcellular localisation and ontological classification of their interacting protein partners. While these data are interesting, the paper would be greatly improved by the direct demonstration of differential subcellular colocalisation of the L1 proteins with their interacting partners. Indeed, the experiments aimed at verifying these conclusions are not entirely convincing, and in some cases the results are overstated or overgeneralised (see below).

2) The authors performed ORF2p-3XFLAG affinity capture from chromatin-enriched subcellular fractions to support the conclusion that ORF2p-only RNPs localise to the nucleus. The text states that proteins co-purifying with chromatin-associated ORF2p "largely overlapped" with those described as nuclear ORF2p-only RNP interactors. However, comparing the list of these proteins in Supplementary file 3 to the list of ORF2-only interactors in Table 1 reveals relatively little overlap. Table 1 contains 21 ORF2p-only interactors and Table S3 contains 15 nuclear ORF2p interactors, yet only five factors (PARP1, IPO7, PCNA, PURA, and TOP1) are actually shared by both lists. The statement that the lists "largely overlapped" needs to be softened, and the extent of the overlap explicitly stated in the text.

3) Immunofluorescence experiments have little quantitative information. Statements like "these cells accounted for <10% of the population" and "Expression of ORF2p in the absence of ORF1p resulted in >90% of cells exhibiting cytoplasmic ORF2p" and "they always occurred in proximal pairs" would be much more convincing with some statistical information about the experiment. How many cells were analysed? How many times were the results replicated? Clarity here would bolster the results.

4) Immunofluorescence experiments could also be used to demonstrate colocalisation of ORF1p and ORF2p with their presumed interaction partners in the cytoplasm and nucleus, and would provide more direct support for the conclusions drawn regarding subcellular localisation of different L1 RNPs. Such experiments would understandably be limited by the availability of antibodies to the interacting factors, but would almost certainly be feasible in some cases.

5) Analysis of host factors differentially associated with RT and EN mutant L1 RNPs is also intriguing, but the results are over-interpreted. As a whole, most of the conclusions drawn in the manuscript need to be backed up by more direct functional assays, or the language in which the conclusions are stated needs to be softened considerably.

*Reviewer #3:*

This is an experimentally sound, well-written manuscript demonstrating the existence of distinct L1 complexes with potentially different roles in L1 retrotransposition. The results presented in this manuscript are novel and provide significant insights into L1 RNPs and its role in the L1 replication cycle. In addition to identifying L1 interactors with a previously reported role in L1 retrotransposition, the authors provide experimental validation of additional and previously unknown interactors. Another important finding is the demonstration that the proportions of proteins within L1 complexes is affected by individual mutations of the L1 ORF2p catalytic activities. I am very enthusiastic about the findings described in this manuscript. I think they significantly advance the research related to transposable elements and viruses and would be helpful to the future progress of studies conducted by other labs.

This reviewer did not have access to the co-submitted manuscript by Mita et al.

• Indicate if the status of L1ORFeus mRNA in the different fractions shown in Figure 1.

• Subsection “4.2. Split-tandem separation of compartment-specific L1 ORF-associated complexes”, the last paragraph provides experimental validation of the significant overlap between putative nuclear L1 complexes and complexes detected in chromatic enriched fractions. Please discuss this validation earlier in this study as it provides important experimental support for the references to these complexes as nuclear.

• Comment on how these nuclear and cytoplasmic complexes align with the existing literature showing that L1 RT present in the cytoplasm generates cDNA when provided with a primer (LEAP protocol).

• "We did not observe instances of nuclear ORF2p using the construct (Figure 2, bottom row), suggesting that ORF1p is required for ORF2p nuclear localization." This statement needs to be put in the context of published data demonstrating that the ORF2p expressing plasmids support efficient Alu retrotransposition.

• In Figure 5, the distance is presented "on a one-unit arbitrary scale." Provide more information on how this unit was developed.

• The model in Figure 6 and several statements throughout the manuscript indicate a complete lack of specific proteins in the nuclear and cytoplasmic complexes. Please clarify if this is an accurate interpretation. Are some proteins enriched in specific complexes and depleted in others rather than absent?

• Statements that are supported by unpublished data that are not shown in this or the accompanying manuscript should be eliminated or restated.

• Discuss the possibility that the lack of detection of the L1 ORF1p in the nucleus by IHC could be due to technical problems rather than the lack of the protein?

---

## [Author Response]

Reviewer #1:This study provides a significant advance in our understanding of biochemical interactions of LINE-1 encoded proteins, with solid evidence for multiple, varied interactions with distinct host cell components in both the nucleus and the cytoplasm. The authors employ sophisticated, quantitative methods to interrogate protein interactions of (primarily) the LINE-1 ORF2 protein, thereby gaining new insight into the complexity of the host-parasite relationship and process of LINE-1 retrotransposition. It represents a significant step forward for the field, building on results of Taylor published in 2013 in Cell. There are however some issues with the wording of some conclusions overstepping the limits of what is documented by the data presented, as specifically noted below. With wording changes to more accurately describe/interpret the data the most serious limitation of this work (the lack of a co-analysis of the L1 nucleic acid intermediates), could be considered minor.

We added LEAP and RNA-seq experiments in order to account for the state of the L1 RNA (and RNA generally) across various fractions – now presented in new Figure 2, and updated Figure 4.

In Abstract "our findings support the presence of multiple L1-derived retrotransposition intermediates in vivo"; oversteps the actual data because there is no direct evidence to address which of these various complexes are bona fide retrotransposition intermediates vs which reflect the various ways the cell interferes with/prevents retrotransposition.

We agree that the diversity of putative assemblies are of unknown molecular physiology (and that their ultimate outcome is unknown), and accordingly, more nuance is required to avoid over-lumping and over-interpretation. We amended the text and title. In numerous places we substituted e.g. “L1 macromolecules” and “protein complexes” in place of “intermediates.” A few instances of “intermediate” were retained where it seemed appropriate for the concept being conveyed. We have also shown the preps to be enzymatically active by LEAP assay (new Figure 2).

In numerous instances the term "purified" is used when the data support "enrichment" but not purification. This leads to confusion when the "purified" complex turns out to be, as is the point of the work, multiple distinct complexes.

Agreed: we have modified numerous instances of “purified,” with alternative descriptors such as “enriched” and “affinity captured.”

The conclusion "[…]ORF1p probably does not interact with ORF2p in the nucleus", is not supported by the data, which simply show that there are RNA- and ORF1p-independent interactions between ORF2p and non-L1 cellular proteins in the nucleus. Agreed that the data show a persistent complex of ORF2p that lacks ORF1p, but that is not the same as the above. There is likewise no data to support "ORF1p may be displaced from the L1 RNA during the nuclear portion of the lifecycle" – maybe there is also no remaining L1 RNA by the time the ORF2 complexes lacking ORF1p are formed. The authors should provide supporting data to support this claim or remove it.

We removed the text “ORF1p probably does not interact with ORF2p in the nucleus" and “ORF1p may be displaced from the L1 RNA during the nuclear portion of the lifecycle.”

Subsection “4.4. The effects of retrotransposition-blocking point mutations on the interactomes of purified L1 RNPs”. TPRT is target primed reverse transcription, not "template".

Fixed.

Subsection “4.4. The effects of retrotransposition-blocking point mutations on the interactomes of purified L1 RNPs”. What evidence suggests that ORF2 traverses L1 RNA in cytoplasm? Given that this RT gets its primer from the target DNA this comment seems unwarranted. This unnecessary speculation is not appropriate for the Results section. In fact the effects of the RT and EN mutations are actually so subtle (~2x) compared to the robust effects they have on retrotransposition (>>100X) it may speak more to the fact that neither of these most abundant L1-containing protein complexes (both nuclear and cytoplasmic) are on the productive retrotransposition pathway.

We removed the speculative text pertaining to ORF2p traversing L1 RNA in cytoplasm.

Figure 6 model – there is no evidence linking the L1 RNA to this nuclear complex with ORF2p and without ORF1p. Given that ORF1p is required to get ORF2p into the nucleus, that some ORF1p does get to the nucleus and in a form that is somehow unique (based on different reactivity of the various antibodies), and that 25% of the ORF2p follows ORF1p in the sequential enrichments, an important role for ORF1p in the nucleus seems highly likely. It would be reasonable from the data presented to consider at least two nuclear forms with ORF2p containing complexes, one with the L1 RNA and ORF1p and one without these other L1 components but with the other nuclear components.

We agree that ORF1p may play a role in the nucleus, but the current data lack granularity that would allow us to define these complexes further. We note that data in the companion manuscript by Mita et al. suggest that any role of ORF1p in the nucleus is transient and show no nuclear co-localization between ORF1p and ORF2p. Accordingly we have modified the figure to show both RNA-linked and non-linked ORF1p in the nucleus, with a semi-transparent shade to imply our lack of knowledge regarding these interactions (now Figure 7).

We now provide two lines of evidence supporting the presence of the L1 RNA in the ORF2p-containing nuclear complex lacking ORF1p: (1) high levels of LEAP activity and (2) RNA sequencing data. We respectfully disagree, therefore, that the ORF1p-lacking complex should also lack the L1 RNA. The remaining schematic accordingly shows ORF2p as the protein interaction hub, in keeping with our lack of evidence to support RNA as a key player in those interactions. We note additionally that the presumed mechanism of L1 insertion, TPRT, would require an ORF2p-L1 RNA interaction at the chromatin. Disruption of e.g. PCNA and UPF1 interactions with ORF2p have strong negative effects on retrotransposition; mild positive effects were observed when knocking down TOP1 and PURA/B; we also identified most of these proteins in ORF2p affinity captured from fractionated chromatin. We conclude that these ORF2p interactions modulate retrotransposition and very likely occur at the chromatin.

Reviewer #2:[…] 1) Most of the conclusions about the subcellular localisation of ORF1p/ORF2p versus ORF2p-only RNP complexes are drawn from the known subcellular localisation and ontological classification of their interacting protein partners. While these data are interesting, the paper would be greatly improved by the direct demonstration of differential subcellular colocalisation of the L1 proteins with their interacting partners. Indeed, the experiments aimed at verifying these conclusions are not entirely convincing, and in some cases the results are overstated or overgeneralised (see below).2) The authors performed ORF2p-3XFLAG affinity capture from chromatin-enriched subcellular fractions to support the conclusion that ORF2p-only RNPs localise to the nucleus. The text states that proteins co-purifying with chromatin-associated ORF2p "largely overlapped" with those described as nuclear ORF2p-only RNP interactors. However, comparing the list of these proteins in Supplementary file 3 to the list of ORF2-only interactors in Table 1 reveals relatively little overlap. Table 1 contains 21 ORF2p-only interactors and Table S3 contains 15 nuclear ORF2p interactors, yet only five factors (PARP1, IPO7, PCNA, PURA, and TOP1) are actually shared by both lists. The statement that the lists "largely overlapped" needs to be softened, and the extent of the overlap explicitly stated in the text.

Fixed. The overlapping proteins have been more explicitly stated in the main text, and the supplementary table has been amended to draw attention to the proteins in question. We purposefully restricted the stated overlap between the proteins identified in the two different experimental designs to those proteins meeting the conservative filter of “I-DIRT specific;” detection alone was not considered sufficient. The rationale is set out in the first two paragraphs of the Results. E.g. the proteins SSBP1, PRPF6, RBM14, U2SURP, SRSF5 (not described as overlapping) were detected in the chromatin fraction ORF2p IP as well as at least one other IP presented in this study, but we do not consider that a compelling enough case to present them as putative L1 interactors at this time. Moreover, we note that in the chromatin fraction ORF2p IP experiment presented in the supplement, of the proteins found to be enriched with ORF2p in the chromatin fraction, only PCNA was also detected after IP of ORF2p from unfractionated cell extracts – this is a general limitation in the sensitivity and dynamic range provided by shotgun proteomic experiments – such that it is not uncommon to observe limited overlap in different proteomic experimental regimes.

3) Immunofluorescence experiments have little quantitative information. Statements like "these cells accounted for <10% of the population" and "Expression of ORF2p in the absence of ORF1p resulted in >90% of cells exhibiting cytoplasmic ORF2p" and "they always occurred in proximal pairs" would be much more convincing with some statistical information about the experiment. How many cells were analysed? How many times were the results replicated? Clarity here would bolster the results.

These images were collected a couple years ago and at the time we did not have the foresight to plan a rigorous quantitative analysis. We counted many cells we observed and estimated frequencies of subpopulations, but did not record images from the population that were not “interesting” with respect to observed ORF1p/ORF2p expression – moreover, aside from the nuclear ORF2p cells, the results were comparable to those we previously published in Taylor et al. 2013. Unfortunately, we currently lack the personnel to revisit this experimental regime. We therefore softened our language regarding the precise proportions of the populations observed. However, to respond to the reviewer and strengthen the manuscript, we added a new companion analysis. We possess 11 such IF images displaying the described ORF2p staining phenomenon, among other non-staining cells (now provided in supplement along with the extracted/analyzed data tables). We analyzed the distributions of cell-cell proximities (nuclei) in these images to provide a quantitative validation of our claim regarding the proximity of cells displaying nuclear ORF2p – now presented in Figure 3. We used a comparable analysis procedure to that presented by Mita et al. in the co-submitted *eLife* manuscript.

4) Immunofluorescence experiments could also be used to demonstrate colocalisation of ORF1p and ORF2p with their presumed interaction partners in the cytoplasm and nucleus, and would provide more direct support for the conclusions drawn regarding subcellular localisation of different L1 RNPs. Such experiments would understandably be limited by the availability of antibodies to the interacting factors, but would almost certainly be feasible in some cases.

We do certainly agree that in some cases these experiments would be feasible and informative. As stated above, there are some expected limitations related to identifying reagents, as well as expectations that some may not generate discrete/enriched co-localization foci. E.g. as annotated in Uniprot, many have broad, diffuse, or complex localizations – we also agree with the reviewer’s point that localization of these proteins in the context of L1 expression may differ from that defined by normal conditions, curated in databases. Hence, this experiment would require some time for trial and error on our part and, given IF is not the expertise of our core-team, was not an experiment we could implement within the post-review timeframe.

We have instead softened our language throughout (in response to this and other reviewer points), in order to compensate for the reviewer’s degree of skepticism regarding putative L1 complexes, and we have added additional IF co-localization as a future experimental direction in the discussion. We also believe this point is at least partly addressed by the chromatin fractionation experiment (pt. 1, above) which showed compositional overlap with our proposed nuclear L1 intermediates inferred from alternative fractionation schemes (Figure 1).

5) Analysis of host factors differentially associated with RT and EN mutant L1 RNPs is also intriguing, but the results are over-interpreted. As a whole, most of the conclusions drawn in the manuscript need to be backed up by more direct functional assays, or the language in which the conclusions are stated needs to be softened considerably.

The manuscript as a whole has been modified to soften tone, including the changes made in response to reviewer #1; in this section we removed the text pertaining to ORF2p traversing L1 RNA in cytoplasm, as well as other modifications.

Reviewer #3:[…] • Indicate if the status of L1ORFeus mRNA in the different fractions shown in Figure 1.

RNA-seq and LEAP analysis of the fractions has now been provided, see new Figure 2.

• Subsection “4.2. Split-tandem separation of compartment-specific L1 ORF-associated complexes”, the last paragraph provides experimental validation of the significant overlap between putative nuclear L1 complexes and complexes detected in chromatic enriched fractions. Please discuss this validation earlier in this study as it provides important experimental support for the references to these complexes as nuclear.

We attempted to accommodate this critique, however, we cannot move this section to an early position in the manuscript because the section, as written, leverages the observation of TOP1 as part of the putative nuclear L1 complex. TOP1 was not detected in the RNase experiment, and therefore, moving this section requires some acrobatics to narrate the significance of the TOP1 observation and testing of TOP1 by over-expression, etc. If the reviewer is adamant on this request, we could accommodate it with a more involved re-write where we excise the text describing the chromatin fractionation and move it to the previous section, but leave the over-expression results where they are (following the observation of TOP1). We note that, as it stands, this information is provided in the Results section attached to Figure 1, so, it is provided at a reasonably early juncture.

• Comment on how these nuclear and cytoplasmic complexes align with the existing literature showing that L1 RT present in the cytoplasm generates cDNA when provided with a primer (LEAP protocol).

We have done LEAP experiments on split-tandem affinity captured L1 fractions (those of Figure 1), and all fractions are highly active (new Figure 2). The absolute activity between the fractions is similar (within 2-fold for LD401) but the RNA and protein partitioning is not, and thus depending on how the data is normalized (to ORF2p or the L1 RNA) the specific activity can be viewed in different ways. These experiments require further investigation and the data will require a thorough and nuanced discussion that will be included in a follow-up study we’re conducting on the enzymatic activities of purified L1 macromolecules. For now, we can conclude that the RNA-ORF2p complex seems competent to perform LEAP given the appropriate primer, with or without detectable ORF1p. Further we would note that the precise subcellular localization of the active L1 RT in previously reported LEAP preparations is far from clear. Our prior work was whole cell extracts made by cryomilling; the Moran lab protocol uses roughly 3 packed cell volumes of a hypotonic buffer with 1% Triton X-100 and 1% deoxycholate detergents before ultracentrifugation and contains protein from multiple cellular compartments, along with ~65-fold less specific activity than the affinity-purified fractions. Thus at least formally, we don’t know that the subcellular origin of LEAP-competent RT in these assays.

• "We did not observe instances of nuclear ORF2p using the construct (Figure 2, bottom row), suggesting that ORF1p is required for ORF2p nuclear localization." This statement needs to be put in the context of published data demonstrating that the ORF2p expressing plasmids support efficient Alu retrotransposition.

This has been addressed in the Discussion.

• In Figure 5, the distance is presented "on a one-unit arbitrary scale." Provide more information on how this unit was developed.

This is explained in the Materials and methods section: “To integrate and plot the combined data (Figure 5 – now Figure 6), we calculated Euclidean and cosine distances for each I-DIRT-significant protein pair present in each experiment. Euclidean distances were rescaled to the range (0, 0.9). Proteins not detected in any common experiments were assigned a Euclidian distance of 1 after rescaling. The total distance between protein pairs was calculated as d = log((rescaled Euclidean distance) * (cosine distance)). This distance was rescaled to the range (0, 1). Hierarchical clustering was used to visualize the calculated distances.” We have added a note to the figure legend referring the reader to this part of the Materials and methods section.

• The model in Figure 6 and several statements throughout the manuscript indicate a complete lack of specific proteins in the nuclear and cytoplasmic complexes. Please clarify if this is an accurate interpretation. Are some proteins enriched in specific complexes and depleted in others rather than absent?

If we understand the reviewer’s question correctly, using the split tandem experiment as a model to answer – only ORF1p was quantitatively shifted at or nearly 100% to the elution fraction of the 2D purification (binary relationship – it is in one fraction and not in another). Note that, for technical reasons related to quantitation using shotgun proteomics, zero-values for proteins (that may actually not be present) are not recorded in any sample, and small values instead are derived either from noise or imputed. ORF1p was shown to be ~100% in the final elution fraction by western blot (Taylor et al. 2013) and by MS (this study). Other proteins generated much of their signal (as graphed) in the ORF1p containing fraction, but some proportion may have decayed away (remaining in the supernatant) or be partially associated with the ORF2p-only fraction. We cannot distinguish between these scenarios based on that data alone. The inverse is also true for proteins in the supernatant. UPF1 represents a protein that we hypothesize to be specifically present in both complexes because of its borderline behavior in numerous experiments. Being an RNP that evolves from one lifecycle stage to the next, proteins likely exchange on and off the complex dynamically along the way. These proteins may exhibit absolute or conditional dependencies, resulting in heterogeneous mixtures or continua of RNPs. Figure 3 (now Figure 4) demonstrates varying degrees of coordination of protein associations with L1 based on catalytic mutants. E.g. association with TOP1 seems nearly-dependent on catalytic activity or an activity-associated cellular context. Figure 4 (now Figure 5) demonstrates varying degrees of coordination in protein decay and exchange from L1s in vitro. Several proteins appear to behave nearly identically, within different classes. However, there are few binary, strictly present/not present, relationships observed across these many experiments. Figure 5 (now Figure 6) is an attempt to synthesize the sum of the data into a model of co-behavioral patterns that explain the results. Figure 6 (now Figure 7) is a schematic that reflects our degree of belief, based on the data, of physiologically relevant steady state features of L1. We cannot rule out that some nuclear proteins that are also found in the cytoplasm could join L1s in the cytoplasm (making no assumptions about a given RNP’s continued progress in the lifecycle), or the vice versa. We assume that some host factors assembled in the cytoplasm may indeed carry through to the nucleus, perhaps UPF1 is an example, or, perhaps UPF1 joins cytoplasmic and nuclear L1 fractions in distinct binding events with distinct outcomes. Dissecting these relationships in greater resolution is the topic of our ongoing work.

• Statements that are supported by unpublished data that are not shown in this or the accompanying manuscript should be eliminated or restated.

Done.

• Discuss the possibility that the lack of detection of the L1 ORF1p in the nucleus by IHC could be due to technical problems rather than the lack of the protein?

This section has been re-written; we do not make a point of not detecting ORF1p in the nucleus in our IF assays anymore, and instead refer to Mita et al. the co-submission that does demonstrate nuclear ORF1p. We understand that this reviewer did not have access to the co-submitted manuscript by Mita et al. – in its current state the text stands on its own and does not suggest a failure to detect nuclear ORF1p means a lack of ORF1p in the nucleus.